# Inclusion of a suite of weathering tracers in the cGENIE Earth System Model - muffin release v.0.9.22

Markus Adloff[1], Andy Ridgwell[1,2], Fanny M. Monteiro[1], Ian J. Parkinson[3], Alexander J. Dickson[4], Philip A. E. Pogge von Strandmann[5,6], Matthew S. Fantle[7], and Sarah E. Greene[8]

[1]BRIDGE (Bristol Research Initiative for the Dynamic Global Environment), School of Geographical Sciences, University of Bristol, Bristol, UK
[2]Department of Earth and Planetary Sciences, University of California Riverside, Riverside, California, USA
[3]School of Earth Sciences, University of Bristol, Bristol, UK
[4]Department of Earth Sciences, Royal Holloway University of London, London, UK
[5]London Geochemistry and Isotope Centre (LOGIC), Institute of Earth and Planetary Sciences, University College London and Birkbeck, University of London, London, UK
[6]Institute of Geosciences, Johannes Gutenberg University, 55122 Mainz, Germany
[7]Department of Geosciences, Penn State University, Pennsylvania, USA
[8]School of Geography, Earth and Environmental Sciences, University of Birmingham, Birmingham, UK

**Correspondence:** Markus Adloff (m.adloff@bham.ac.uk)

**Abstract.** The metals strontium (Sr), lithium (Li), osmium (Os) and calcium (Ca) and their isotopes are important tracers of weathering and volcanism, two main processes which shape the long-term cycling of carbon and other biogeochemically important elements at the Earth's surface. Traditionally, isotopic shifts of these four elements in the geologic record are interpreted with isotope-mixing, tracer-specific box models because of their long residence times in the ocean. However, as such, these models often lack mechanistic links between the cycling of the four metals and other geochemically relevant elements, particularly carbon. Here we develop and evaluate an implementation of Sr, Li, Os and Ca isotope cycling into the Earth system model cGENIE. The model offers the possibility to study these metal systems at equilibrium and under perturbations alongside other biogeochemical cycles. We provide examples of how to apply this new model to investigate Sr, Li, Os and Ca isotope dynamics and responses to environmental change.

## 1 Introduction

The evolution of life and climate on Earth is intrinsically linked to the dynamics of carbon, oxygen and nutrients at the Earth's surface. While complex interactions on a range of spatial and temporal scales determine the distribution of these elements between surficial (non-lithological) reservoirs like oceans, atmosphere, biomass and soils, the total amount and isotopic distribution of these elements are ultimately governed by the balance between crustal weathering, marine sediment deposition and mantle inputs during oceanic crust formation. Geological records show that the abundance of these elements at Earth's surface has varied on time scales of thousands to millions of years. For example, the changing composition of mantle fluxes into the ocean and nutrient supply during weathering of large mountain ranges were potentially drivers of the accumulation of surficial oxygen during the Archean and Proterozoic (e.g. Kump and Barley, 2007; Campbell and Allen, 2008). Likewise,

the supply of particulate iron from airborne grounded rock to the oceans of the Quaternary is invoked as a mechanism to drive glacial-interglacial carbon cycle dynamics (e.g Martin, 1990; Martínez-Garcia et al., 2011; Loveley et al., 2017; Hooper et al., 2019). Increased basalt weathering in the Cenozoic could have led to a continuous decline in $CO_2$ concentrations, contributing to the reconstructed long-term cooling trend (e.g. Kent and Muttoni, 2013; Mills et al., 2014) while the emplacement of large igneous provinces (LIPs) repeatedly led to climatic and biotic crises by increasing the supply of mantle-derived carbon and nutrients to the oceans, e.g. during oceanic anoxic events (e.g. Erba et al., 2010; Jenkyns, 2010; Bottini et al., 2012; Monteiro et al., 2012; Percival et al., 2015).

Mass fluxes between the lithosphere and Earth's dynamic surface are thus important drivers of environmental processes on the Earth's surface over geological time scales, but our ability to measure these fluxes directly is limited by their long evolution times and large (e.g. riverine) spatial heterogeneity. Instead, records of variations in the oceanic content and isotopic composition of trace metals like strontium (Sr), osmium (Os), lithium (Li) and calcium (Ca), preserved in marine sediments, are used to study lithological processes. Although the dynamics of these metals are controlled by processes that also shape the long-term carbon and nutrient cycles, namely continental and oceanic weathering and direct mantle emissions, variations in their isotopic compositions can be linked more directly to changes in these processes since radioactive decay and mass-dependent fractionation form isotopically-distinct lithological metal reservoirs. Furthermore, the long residence time of these metals in the ocean, which is generally assumed to be greater than the mixing time of the ocean, removes the complexity of spatially heterogeneous metal sources from marine records since their marine distribution is comparably homogeneous (e.g. Faure and Mensing, 2005) and marine sedimentary records are typically understood as capturing global signals.

Despite laboratory methods to measure and reconstruct the evolving trace metal composition of seawater have becoming increasingly precise, our ability to interpret these records is limited by our incomplete mechanistic understanding of the cycling of these metals and their isotopes. Quantifying reservoir sizes and fluxes constitute particular challenges under under different biogeochemical, climate, and palaeogeographic background states and marine productivity over geological time. Mass balances and box models have been most commonly used to interpret geological records of trace metal variations (e.g. Tejada et al., 2009; Misra and Froelich, 2012; Kristall et al., 2017; Them et al., 2017). They cannot mechanistically resolve spatially heterogeneous metal burial and hence often rely on assumed global fluxes and residence times based on today's ocean which might not always be applicable throughout the geological record. Additionally, they do not include effects of Earth system feedbacks (e.g. climate-driven weathering flux changes, sediment dissolution), the simulation of which requires information about spatially heterogeneous climate and ocean variables, bathymetry and continental configuration. In contrast, more mechanistic and spatially-resolved Earth system Models of Intermediate Complexity ('EMIC') provide an alternative approach. In particular, the EMIC 'cGENIE' has been used successfully to study marine biogeochemical cycles (including C and Ca) under various boundary conditions and external forcings (e.g. Ridgwell and Zeebe, 2005; Ridgwell et al., 2007; John et al., 2014; Death et al., 2014; Turner and Ridgwell, 2016; Hülse et al., 2019). Implementing isotope enabled cycling of Sr, Os, Li and Ca in cGENIE allows us to test their behaviour under environmental perturbations alongside nutrient, redox and carbon dynamics and compute their spatial distributions and marine residence times with different assumptions about source and sink mechanisms .

Here, we present an implementation of the marine cycling of Sr, Os, Li and Ca in cGENIE. We evaluate the model's performance by comparing simulations of pre-industrial trace metal distributions in the oceans to seawater measurements, and study their equilibration times under imposed geochemical perturbations compared to established seawater residence times.

## 2 Observational constraints

We first provide a brief review of Sr, Os, Li and Ca cycling in the ocean (sections 2.1 - 2.4), as well as their observed concentrations and isotopic compositions in seawater (2.5) and their applications as proxies for Earth system processes (2.6). This discussion provides the theoretical basis for our model implementation, and the data compilations for model evaluation at the end of this study.

### 2.1 Strontium

Of the four metal cycles simulated in this study, Sr has the second highest marine concentration (3 – 4 times more abundant than Li and more than one billion times more abundant than Os, Broecker and Peng, 1982). The main source for marine Sr is continental weathering, replenishing the ocean reservoir through rivers and potentially groundwater discharge (Basu et al., 2001; Beck et al., 2013). Smaller sources include hydrothermal input of mantle-derived Sr and refluxes from diagenetic alteration of carbonates at the sea floor. The chemical and physical similarity of Sr to Ca leads to its substitution in aragonite

and, to a lesser degree, calcite (Fietzke and Eisenhauer, 2006; Rüggeberg et al., 2008; Böhm et al., 2012; Stevenson et al., 2014). Angino et al. (1966) argue that biological uptake is the dominant driver of spatial gradients in Sr concentrations in today's oceans, and Krabbenhöft et al. (2010) predict that marine carbonate burial could be the most important sink of seawater Sr. The concentration of Sr incorporated into biogenic carbonates depends on the mineralogy and the growth rate with higher Sr/Ca ratios in aragonite precipitated at fast growth rates (Rickaby et al., 2002; Stevenson et al., 2014). By contrast, direct effects

of temperature on elemental fractionation during the formation of calcite, the major carbonate buried in marine sediments, are small (Tang et al., 2008). Assuming rivers to be the only supply of continental Sr to the oceans and that the marine Sr reservoir size is in equilibrium, Hodell et al. (1989) estimates the marine residence time of Sr to be 1.9 – 3.45 Myr. Considering the potentially appreciable Sr influx from groundwater, the actual residence time might be at or below the lower end of this estimate, or yet different if the marine Sr reservoir is not currently in equilibrium with Sr inputs as suggested by e.g. Vance

et al. (2009).

  Strontium has four stable isotopes (0.56% $^{84}$Sr, 9.87% $^{86}$Sr, 7.04% $^{87}$Sr, 82.53% $^{88}$Sr) (Veizer, 1989), although $^{87}$Sr is also the product of $\beta$-decay of $^{87}$Rb (half-life $4.88 \cdot 10^{10}$ yr, Faure and Mensing, 2005). Radiogenic Sr isotope ratios are reported as $^{87}$Sr/$^{86}$Sr ratios whereas stable Sr isotope ratios are reported as $\delta^{88/86}$Sr, which is the per mille (‰) deviation in the $^{88}$Sr/$^{86}$Sr relative to the NBS987 standard ($\delta^{88/86}$Sr $= (\frac{(^{88}\text{Sr}/^{86}\text{Sr})_{sample}}{(^{88}\text{Sr}/^{86}\text{Sr})_{std}} - 1) \cdot 1000$, Fietzke and Eisenhauer, 2006). Ele-

mental fractionation during magmatic processes creates different Rb/Sr ratios that over time generate reservoirs with distinct Sr isotopic signatures, with the continental crust being considerably more radiogenic than the mantle (Faure and Mensing, 2005). Dust and rainwater isotopic signatures, both radiogenic and stable, are generally lower than modern day seawater values

(Pearce et al., 2015). Volcanic material brings unradiogenic particulate Sr into sediments and can over time significantly affect the radiogenic composition of dissolved Sr in the sediment column while having little effect on the isotopic composition of dissolved Sr in seawater (Elderfield and Gieskes, 1982; Pearce et al., 2015). Changes in isotope abundances caused by kinetic and equilibrium mass-dependent fractionation during carbonate formation are much smaller, and one can thus assume that the

$^{87}$Sr/$^{86}$Sr of seawater and deposited carbonates is only controlled by inputs to the ocean (Krabbenhöft, 2011). By contrast, variations in $\delta^{88/86}$Sr are only controlled by fractionation during chemical processes, biogenic carbonate formation being the most important one in the ocean (e.g. Krabbenhöft et al., 2010). There is no evidence for a general dependence of $\delta^{88/86}$Sr fractionation during biogenic carbonate formation on environmental conditions, although temperature and growth rate effects were observed in some calcifying species (Fietzke and Eisenhauer, 2006; Böhm et al., 2012; Stevenson et al., 2014; Vollstaedt

et al., 2014).

**Table 1.** Estimates of pre-industrial Sr fluxes between surface reservoirs and their isotopic composition.

| Process | Flux (Gmol/yr) | $^{87}$Sr/$^{86}$Sr | $\delta^{88/86}$Sr (‰) | Reference |
|---|---|---|---|---|
| hydrothermal input flux | 3-4 | 0.7035-0.70387 | 0.328-0.422 | Pearce et al. (2015); Kristall et al. (2017) |
| input from diagenesis | 3-5 | 0.7035-0.7084 | 0.27 | Kristall et al. (2017) |
| dissolved riverine input | 20.2-47 | 0.7111-0.7136 | 0.32 | Allègre et al. (2010) <br> Peucker-Ehrenbrink et al. (2010) <br> Pearce et al. (2015); Kristall et al. (2017) |
| particulate riverine input | 5.2 | <0.7136 | uncertain | Allègre et al. (2010) <br> Peucker-Ehrenbrink et al. (2010) <br> Kristall et al. (2017) |
| groundwater discharge | 7.1-16.6 | 0.7089 | $0.354 \pm 0.028$ | Basu et al. (2001); Beck et al. (2013) |
| dust flux and rainwater | uncertain | 0.7075 - 0.7191 | 0.05 - 0.31 | Pearce et al. (2015) |
| pelagic Sr burial in carbonates | 12.5 - 174 | seawater | 0.20 | Krabbenhöft et al. (2010) <br> Stevenson et al. (2014) <br> Kristall et al. (2017) |
| neritic Sr burial in carbonates | 19 | seawater | 0.21 | Krabbenhöft et al. (2010) |
| Sr burial in sea floor alteration | uncertain | seawater | seawater | Menzies and Seyfried Jr (1979) <br> Kristall et al. (2017) |

## 2.2   Osmium

Osmium, a siderophile and chalcophile element, is compatible during mantle melting and as such accumulated in Earth's core (Goldschmidt, 1922). As a result, it is one of the rarest elements in the Earth's crust and the oceans. Because of its low concentrations, the ability to measure Os in seawater only developed in the past few decades, limiting the number of

observations of marine Os concentrations and isotopic compositions.

The cycling of Os at the Earth's surface is similar to that of radiogenic Sr in many aspects. Oxidative weathering of sediments is currently the most important natural source of Os to the oceans (Peucker-Ehrenbrink and Ravizza, 2000; Lu et al., 2017). The estimated annual input of such Os to the oceans via aeolian and riverine transport and groundwater discharge is about 5 times larger than all other natural inputs combined (Lu et al., 2017). Hydrothermal inputs constitute the next largest source

of oceanic Os with high and low temperature systems being of roughly equal importance (Georg et al., 2013). The high temperature hydrothermal source can be split into a basaltic and a peridotitic source (Burton et al., 2010). Unlike for Sr, cosmic and terrestrial dust are significant sources of marine Os (Sharma et al., 2007). Os is removed from the ocean by deposition at the seafloor, predominantly under suboxic conditions and in association with organic matter, and in ferro-manganese nodules (Lu et al., 2017). Os incorporation into biogenic carbonates constitutes an additional but minor sink (Burton et al., 2010). The

marine residence time of Os is inherently uncertain given the uncertainty in Os fluxes, although estimates range from 3 – 50 kyr (Sharma et al., 1997; Levasseur et al., 1998; Oxburgh, 2001).

Os has seven naturally occurring isotopes (0.02% $^{184}$Os, 1.59% $^{186}$Os, 1.51% $^{187}$Os, 13.29% $^{188}$Os, 16.22% $^{189}$Os, 26.38% $^{190}$Os, 40.98% $^{192}$Os). $^{186}$Os has such a long half-life that it can also be treated as stable over geologic time, whereas $^{187}$Os is radiogenic due to the beta decay of $^{187}$Re (Faure and Mensing, 2005). The isotopic composition of Os in marine sediments

provides insight into changing fluxes between different Os reservoirs, particularly weathering related Os fluxes from the continents and mantle derived fluxes from volcanic activity. During partial melt in the upper mantle Os is compatible while rhenium (Re) is incompatible, leading to an increased Re/Os ratio in continental crust relative to the mantle (Dąbek and Halas, 2007) and therefore elevated $^{187}$Os/$^{188}$Os ratios in the continental crust compared to the mantle. No isotopic fractionation has been observed during Os uptake into macroalgae (Racionero-Gómez et al., 2017), nor is it assumed to occur during other transfers

between surface reservoirs.

**Table 2.** Estimates of pre-industrial Os fluxes between surface reservoirs and their isotopic composition. In the calculation of Os burial with Ca carbonate, the estimates of Ca burial in table 4 were used.

| Process | Flux (mol/yr) | $^{187}$Os/$^{188}$Os | Reference |
|---|---|---|---|
| riverine input (corrected for estuaries) | 1404-1493 | 1.2-1.5 | Sharma et al. (2007); Georg et al. (2013) |
| groundwater discharge | 957 | 1.2-1.5 | Lu et al. (2017) |
| terrogenous and cosmic dust | 184-463 | 0.12-0.60 | Lu et al. (2017) |
| high-temperature hydrothermal input | 158 | 0.13 | Sharma et al. (2007); Georg et al. (2013) |
| low-temperature hydrothermal input | 100-294 | 0.88 | Sharma et al. (2007); Georg et al. (2013) |
| burial in biogenic carbonates | 96-169 | seawater | Burton et al. (2010) |
| oxic burial in marine sediments | 11-1956 | seawater | Lu et al. (2017) |
| suboxic burial in marine sediments | 2408-14730 | seawater | Lu et al. (2017) |

## 2.3 Lithium

Lithium is the 25th most abundant element in the Earth's crust, and it is more concentrated in continental crust than in oceanic crust (Baskaran, 2011). Weathering, in particular of silicate rocks, releases dissolved Li to rivers and soils where it is partially removed and bound during clay formation (Kısakűrek et al., 2005; Dellinger et al., 2015; Pogge von Strandmann et al., 2017). The remaining dissolved Li is transported to the oceans via rivers and groundwater. Aeolian transport only contributes a minor flux of Li to the oceans (Baskaran, 2011). Li is also added to the oceans by hydrothermal vents and submarine weathering, but the size of this flux is more uncertain (Chan et al., 1992; Hathorne and James, 2006). Removal from the ocean happens predominantly via Li adsorption onto clay minerals, with a minor proportion is buried with Li-containing biogenic calcite (Hathorne and James, 2006). These removal fluxes are dependent on the abundance of inorganic carbon and pH (Hall and Chan, 2004; Marriott et al., 2004), and are not spatially uniform (Hathorne and James, 2006). The residence time of Li in the ocean is estimated to be 0.3 – 3 Ma (Stoffyn-Egli and Mackenzie, 1984) with more recent estimates closer to 1 Ma (Vigier and Goddéris, 2015).

Li has two stable isotopes (7.52% $^6$Li and 92.48% $^7$Li, Penniston-Dorland et al., 2017) and fractionation processes partition these isotopes between reservoirs at Earth's surface. The isotopic composition of Li is expressed as $\delta^7$Li, which is the ‰ deviation of the $^7$Li/$^6$Li ratio from the L-SVEC standard ($\delta^7$Li = $(\frac{(^7\text{Li}/^6\text{Li})_{sample}}{(^7\text{Li}/^6\text{Li})_{std}}$ - 1) $\cdot$ 1000). While earlier studies suggested considerable spatial variability in seawater $\delta^7$Li (Carignan et al., 2004), more recent studies suggest that seawater is remarkably homogeneous in its $\delta^7$Li (Hall et al., 2005; Rosner et al., 2007; Penniston-Dorland et al., 2017) consistent with its long residence time. Isotope fractionation has been observed during clay formation, adsorption onto minerals and incorporation of Li into calcite shells (Pistiner and Henderson, 2003; Rudnick et al., 2004; Dellinger et al., 2015; Hindshaw et al., 2019), in magmatic systems due to diffusion (Parkinson et al., 2007; Penniston-Dorland et al., 2017), as well as potentially also in aqueous solutions (Richter et al., 2006). Isotopic differences in weathered lithologies and post-weathering formation of secondary minerals in rivers and soils result in large spatial and temporal variability of the $\delta^7$Li of continental run-off (Huh et al., 1998; Pistiner and Henderson, 2003; Dellinger et al., 2015; Pogge von Strandmann and Henderson, 2015). Temporal variations in the amount of secondary mineral formation on land have the potential to drive shifts in seawater $\delta^7$Li over geological time (Misra and Froelich, 2012; Pogge von Strandmann and Henderson, 2015). Fractionation during biogenic carbonate formation results in carbonate $\delta^7$Li values that are a few ‰ lower than seawater, but this offset seems to be carbonate producer-dependent Hathorne and James (2006).

**Table 3.** Estimates of pre-industrial Li fluxes between surface reservoirs and their isotopic composition. In the calculation of Li burial with calcium carbonate, the estimates of calcium burial in table 4 were used, assuming a 50:50 split in calcium carbonate burial fluxes between neritic and pelagic environments (Milliman, 1993).

| Process | Flux (Gmol yr$^{-1}$) | $\delta^7$Li (‰) | Reference |
|---|---|---|---|
| continental run-off | 8-16 | 23 | Hathorne and James (2006); Misra and Froelich (2012) |
| hydrothermal vents | 3-15 | 8.3 | Hathorne and James (2006); Misra and Froelich (2012) |
| subduction reflux | 6 | 15 | Misra and Froelich (2012) |
| loss to sea spray | 0.1 | sea water | Stoffyn-Egli and Mackenzie (1984) |
| secondary mineral formation | 3.5-37 | 16 | Hathorne and James (2006); Misra and Froelich (2012) |
| sea floor alteration | 1-12 | 16 | Hathorne and James (2006); Misra and Froelich (2012) |
| neritic carbonate burial | 0.02 - 1.30 | 20 - 40 | Rollion-Bard et al. (2009); Dellinger et al. (2018) |
| pelagic carbonate burial | 0.12 - 0.23 | 27.1 - 31.4 | Hathorne and James (2006) |

## 2.4 Calcium

Calcium cycling is closely linked to the C cycle, both shaping and shaped by the size of C reservoirs and C fluxes at the Earth's surface. Similar to Sr, Os and Li, the dominant Ca source for today's oceans is weathering-derived dissolved and particulate Ca in continental runoff. Input through hydrothermal vents near ocean ridges is on the order of 20% of the riverine flux (e.g. Milliman, 1993; DePaolo, 2004). Unlike Sr, Os, and Li, Ca plays an important role in many biological systems, predominantly as an electrolyte and building block for biogenic minerals (e.g. shells, exoskeletons, bones and teeth). In the ocean, Ca ions are incorporated into biogenic minerals (e.g. foraminiferal tests and calcareous nannoplankton), or form hydrogenetic or authigenic minerals (Fantle et al., 2020) if waters are highly saturated ($\Omega = a_{Ca^{2+}} \cdot a_{CO_3^{2-}}/K_{sp}$). The resulting minerals are dissolved in undersaturated conditions, or buried, compacted, and lithified. Carbonate formation creates the biggest long-term Ca and C sinks in today's oceans, and marine carbonate accumulation and dissolution constitute a significant buffer mechanism to stabilize marine and atmospheric $p$CO$_2$ during periods of enhanced exogenic C input. The residence time of Ca in seawater is estimated to be 0.5-1.3 million years (Milliman, 1993; Sime et al., 2007; Griffith et al., 2008).

Ca has 6 stable isotopes (96.941% $^{40}$Ca, 0.647% $^{42}$Ca, 0.135% $^{43}$Ca, 2.086% $^{44}$Ca, 0.004% $^{46}$Ca, 0.187% $^{48}$Ca). $^{48}$Ca has such a long half-life that it can be treated as a stable isotope, whereas a small component of the $^{40}$Ca in a rock mineral is radiogenic (via decay of $^{40}$K; $t_1/2$= 1.248 Ga) and accumulates in continental crust over geological time scales (Fantle and Tipper, 2014). The isotopic composition of Ca is reported as either $\delta^{44/40}$Ca or $\delta^{44/42}$Ca, which are the ‰ deviations in $\delta^{44/40}$Ca and $\delta^{44/42}$Ca, respectively, from NIST SRM-915a, SRM-915b, or modern seawater (see Fantle and Tipper, 2014, for discussion): ($\delta^{44/40}$Ca = $(\frac{(^{44}Ca/^{40}Ca)_{sample}}{(^{44}Ca/^{40}Ca)_{std}}$ - 1) $\cdot$ 1000 and $\delta^{44/42}$Ca = $(\frac{(^{44}Ca/^{42}Ca)_{sample}}{(^{44}Ca/^{42}Ca)_{std}}$ - 1) $\cdot$ 1000). We use the $\delta^{44/40}$Ca notation in this study, from now on shortened to $\delta^{44}$Ca, with NIST SRM-915a as standard. Dissolution of carbonates and silicates, as well as the precipitation of secondary silicate minerals, control the Ca isotopic composition in soil pore fluids, lakes and rivers (Farkaš et al., 2007; Tipper et al., 2008; Hindshaw et al., 2013; Fantle and Tipper, 2014; Kasemann et al.,

2014; Perez-Fernandez et al., 2017). Both, biotic and abiotic precipitation of carbonates fractionate Ca isotopically, generating minerals with low $\delta^{44}$Ca values relative to aqueous $Ca^{2+}$. The isotopic fractionation factor is most strongly a function of precipitation rate (and thus and solution chemistry, e.g. DePaolo, 2011) and is close to one ($\Delta \sim 0$) in the marine sedimentary column (see reviews in Blättler et al., 2012; Fantle and Tipper, 2014). In the ocean, species-dependent fractionation has been observed for several groups of calcifiers, with a small dependence on temperature (e.g. Nägler et al., 2000).

**Table 4.** Estimates of pre-industrial Ca fluxes between surface reservoirs and their isotopic composition. $\delta^{44/40}$Ca are given relative to NIST SRM915a (offset by +1.8825±0.07 from seawater (Holmden et al., 2012))

| Process | Flux (Tmol yr$^{-1}$) | $\delta^{44/40}$Ca (‰) | Reference |
|---|---|---|---|
| hydrothermal input flux | 2-20 | 0.93±0.05 | Berner and Berner (2012); Zhu and Macdougall (1998) Holmden et al. (2012); Tipper et al. (2016) |
| input from diagenesis | 0.92 | 0.6±0.77 | Berner and Berner (2012); Fantle and Tipper (2014) |
| riverine input | 13.72 | 0.88±0.5 | Berner and Berner (2012); Holmden et al. (2012) Fantle and Tipper (2014) |
| groundwater discharge | 5.24 - 13.22 | 0.58 - 0.85±0.23 | Berner and Berner (2012); Holmden et al. (2012) Fantle and Tipper (2014) |
| dust flux | 0.05 - 2.25 | 0.72±0.6 | Fantle et al. (2012); Fantle and Tipper (2014) |
| Ca burial in carbonates | 23.95 - 31.94 | 0.58 - 0.78 | Holmden et al. (2012); Fantle and Tipper (2014) |

## 2.5 Metal distributions in seawater

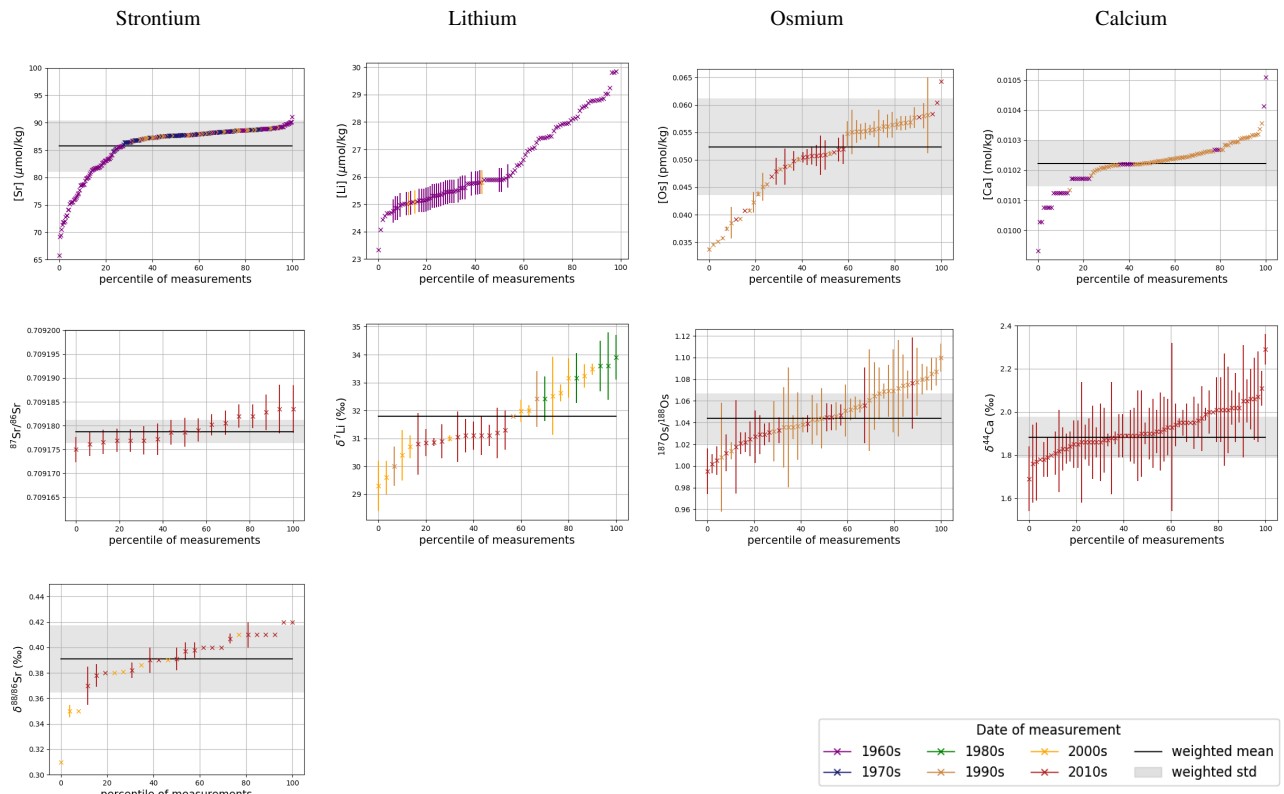

**Figure 1.** Sr, Li, Os and Ca composition of modern seawater. Shown are composites of all published measured concentrations and isotope ratios with reported errors. From these, we calculated mean concentrations and isotope ratios weighted by the reported error (the more uncertain a value, the less it contributes to the mean), which are indicated by horizontal lines. Shading indicates one weighted standard deviation around the means. Data are taken from Angino et al. (1966), Fabricand et al. (1967), Bernat et al. (1972), Brass and Turekian (1974), De Villiers (1999), Pearce et al. (2015), Mokadem et al. (2015) and the compilation of Wakaki et al. (2017) for Sr, Angino and Billings (1966), Fabricand et al. (1967), Chan (1987), Chan and Edmond (1988), You and Chan (1996), Moriguti and Nakamura (1998), Tomascak et al. (1999), James and Palmer (2000), Košler et al. (2001), Nishio and Nakai (2002), Hall (2002), Bryant et al. (2003), Pistiner and Henderson (2003), Millot et al. (2004), Choi et al. (2010), Pogge von Strandmann et al. (2010), Phan et al. (2016), Lin et al. (2016), Henchiri et al. (2016), Weynell et al. (2017), Fries et al. (2019), Gou et al. (2019), Hindshaw et al. (2019), Murphy et al. (2019) and Pogge von Strandmann et al. (2019) for Li, Levasseur et al. (1998), Woodhouse et al. (1999) and Gannoun and Burton (2014) for Os and Fabricand et al. (1967), De Villiers (1999) and Fantle and Tipper (2014) for Ca. The decade of publication is indicated by colour as measurement techniques for some metals/isotopes have improved over time. Concentrations of Sr, Li and Ca are normalized to a salinity of 34.903495. Os concentrations could not be salinity-normalized because of missing salinity measurements at one out of three measurement sites (Woodhouse et al., 1999).

Dissolved Sr, Os, Li and Ca are largely homogeneous in sea water, which is illustrated in figure 1 by sorting all seawater measurements - independent of location and depth - by their measured value. Measurements of a perfectly homogeneous - salinity normalized - seawater property would appear as a horizontal line in this plot, since the same value would be measured everywhere in the ocean. Most trace metal measurements have a difference of less than one standard deviation from the respec-

tive mean, suggesting very small spatial heterogeneities. In particular, most measured isotopic compositions are analytically indistinguishable from the standard deviation of the overall data population. Larger differences between lowest and highest measured metal concentrations indicate heterogeneity (potentially horizontal and/or vertical gradients) in metal abundances or are an artefact of the small number of sub-surface measurements. One possible example is Sr, which is reportedly less abundant in surface waters and waters in the North Atlantic than in deeper waters and sites outside the North Atlantic (see fig. B1).

Higher Li concentrations in the surface ocean compared with deeper waters are only reported in one study, Angino and Billings (1966), while no vertical gradients are apparent in Fabricand et al. (1967) and Hall (2002). These differences suggest that the spread in Li concentrations in fig. 1 could be the result of analytical uncertainty rather than real gradients in dissolved Li. Os concentrations are not salinity-normalized but salinity is unlikely to be the dominant driver of the observed heterogeneity because the relative spread of Os concentrations (factor 2 between the lowest and highest measurements) is larger than that of

salinity (maximum factor of 1.1 Talley, 2002). Instead, there is considerable variation in Os concentration between sites and measurement techniques (Gannoun and Burton, 2014). Ca values show no spatial heterogeneity.

## 2.6  Application of Sr, Os, Li and Ca isotopes as proxies of weathering and mantle emissions

The isotopic differences between Sr, Os, Li and Ca in the ocean and exogenic reservoirs makes these metals useful proxies for mass exchange between the surficial Earth system and the mantle, continental crust (all four) and extra-terrestrial material (Os).

However, they record different geochemical pathways because they behave differently in water and have different predominant host lithologies.

Radiogenic Sr and Os isotopes are used separately as proxies for the balance between the weathering of old and juvenile basalt or direct mantle emissions (e.g. Hodell et al., 1990; Goddéris and François, 1995; Tejada et al., 2009; Finlay et al., 2010; Bottini et al., 2012; Dickson et al., 2015) but in combination, they also provide information about sedimentary rock

weathering since continental Sr is primarily derived from carbonates while Os resides predominantly in shales and evaporites (e.g. Peucker-Ehrenbrink et al., 1995). Li is orders of magnitude more abundant in silicates than in carbonates and thus is discussed as the most direct proxy for silicate weathering (Kısakűrek et al., 2005; Millot et al., 2010; Pogge von Strandmann et al., 2020). The isotopic composition of dissolved Li is also not affected by plant growth (Lemarchand et al., 2010; Clergue et al., 2015) or phytoplankton growth (Pogge von Strandmann et al., 2016). Instead, the light Li isotope ($^6$Li) is preferentially

taken up by secondary minerals (clays, oxides, zeolites) formed during weathering, enriching residual waters in isotopically heavy Li. Hence, surface water $\delta^7$Li is controlled by the ratio of primary rock dissolution to secondary mineral formation, known as the weathering congruency (Misra and Froelich, 2012; Pogge von Strandmann and Henderson, 2015), which in turn can act as a tracer of weathering intensity (that is, the ratio of the weathering rate to the denudation rate, Dellinger et al., 2015; Pogge von Strandmann et al., 2017; Murphy et al., 2019; Gou et al., 2019). This also relates to the efficiency

of $CO_2$ drawdown, as cations retained on the continents in secondary minerals do not end up in the oceans to assist carbon sequestration (Pogge von Strandmann and Henderson, 2015; Pogge von Strandmann et al., 2017). Ca isotopes have also been used to examine weathering processes in the geological record (Kasemann et al., 2005; Farkaš et al., 2007; Kasemann et al., 2008; Blättler et al., 2011; Holmden et al., 2012; Fantle and Tipper, 2014; Kasemann et al., 2014), and additionally are crucial proxies for the quantification of carbonate precipitation rates (Pogge von Strandmann et al., 2019), including authigenesis (Fantle and Ridgwell, 2020). Because of its tight links with the marine carbonate system, Ca also serves as a constraint on $CO_2$ concentration. Furthermore, Ca isotopes have been considered as a potential temperature proxy (e.g. Nägler et al., 2000), but this application might be complicated by the strong control of aqueous chemistry on Ca fractionation (DePaolo, 2011). Alongside Ca, stable Sr isotopes provide information about the dominant form of carbonate burial (Paytan et al., 2021) and growth rate in marine calcifiers, mostly influenced by temperature and $pCO_2$ (Stevenson et al., 2014; Müller et al., 2018). Likewise, changes in the Sr/Ca ratio in calcifiers reflect shifts in ecosystem structure, carbonate mineralogy and calcification rate (Stoll and Schrag, 2001).

While each system individually gives valuable insight into Earth system dynamics, in concert, these trace metal systems can provide information on feedbacks and event durations, as well as improve the accuracy of our reconstructions (e.g. Kasemann et al., 2008).

## 3  Model implementation

We developed parameterizations of marine cycling of Sr, Os and Li and Ca isotopes in the Earth system model 'cGENIE'. This implementation contains the elemental and isotopic fluxes that are most relevant to simulate changes of marine metal reservoirs due to external metal additions or weathering rate changes. The representation of these fluxes can be improved by future model development, particularly on processes occurring in terrestrial freshwater environments and at the seafloor. Section 5 discusses ideas for added complexity. The present section provides an overview of the existing implementation.

cGENIE is best described as a modelling framework (Lenton et al., 2007) and comprises in our study, modules for ocean physics ('GOLDSTEIN' Edwards and Marsh, 2005), marine biogeochemistry ('BIOGEM' Ridgwell et al., 2007), continental weathering and run-off ('ROKGEM' Colbourn et al., 2013), sea-floor sediment formation ('SEDGEM' Ridgwell and Hargreaves, 2007), atmospheric chemistry ('ATCHEM') and atmospheric energy balance ('EMBM' Ridgwell et al., 2007). As such, our implementation of cGENIE includes a 3D ocean combined with a 2D atmosphere and can capture the cycling of carbon and a range of other elements relevant for biogeochemical studies of the ocean water column and at the sea-sediment and sea-air interfaces, as well as climate-sensitive continental run-off and explicit sedimentary carbonate burial.

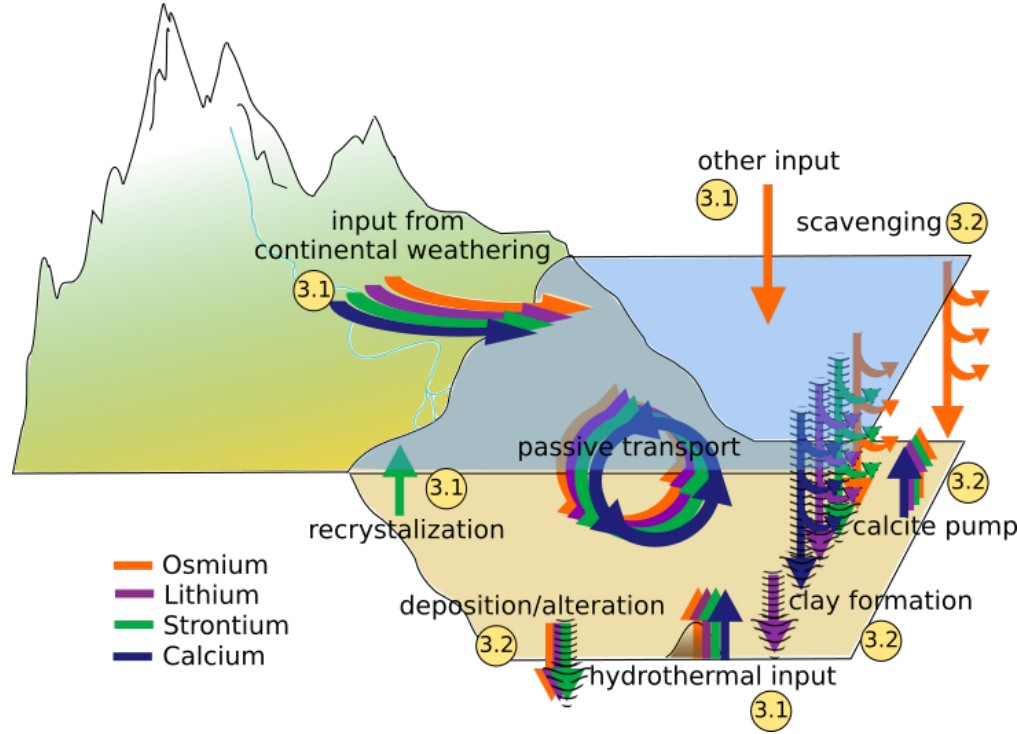

**Figure 2.** Processes included in the cGENIE trace metal implementation and the number of the subsection describing their implementation. Hashed arrows indicate processes during which isotopic fractionation occurs in the model.

Figure 2 shows a conceptual model of the Sr, Os, Li and Ca cycles in cGENIE with arrows of different colours showing mass fluxes of different metals. The implementation of these processes involves code additions to the modules ROKGEM, BIOGEM and SEDGEM that are described in the following sections. Each code addition requires the model user to set parameters controlling the respective metal fluxes. These parameters, as well a parameter set which reproduces the present-day marine distributions of Sr, Os, Li and Ca in steady state metal cycles, are provided in the SI (E).

### 3.1 Marine Sr, Os, Li and Ca sources

Continental weathering is the main source for marine Sr, Os, Li and Ca today and changes in the weathered lithology or weathering intensity are invoked as major drivers of the evolution of trace metal concentrations and their isotopic compositions in seawater over time (e.g. Misra and Froelich, 2012; Kristall et al., 2017). ROKGEM, the weathering module of cGENIE, provides a framework for calculating climate- and $CO_2$-dependent additions of Ca, Mg and alkalinity from carbonate weathering (following Berner, 1994) and silicate weathering (following Brady, 1991) to the ocean (see Colbourn et al., 2013, for a full description of the weathering module). By applying linear and exponential relationships of temperature and carbonate and silicate weathering rates, respectively, ROKGEM modifies user-given baseline weathering fluxes according to climatic variations.

Additional (optional) modifiers can be selected to represent the effect of changes in precipitation, vegetation and atmospheric $CO_2$ concentration.

The modification of weathering fluxes is either calculated locally based on a geographic distribution of carbonates and silicates taken from user-input ('2-D case', described in more detail in the SI) or globally based on global average climate change ('0-D case'). In each case, the module calculates a drainage map based on a prescribed continental topography and determines the coastal locations where weathering input (either of regionally variant composition in the 2-D case or homogeneous in the 0-D case) is added to the ocean. The flux of metals added to the ocean in these locations is determined by the size of the model catchment, and can optionally be scaled by the proportion of global run-off entering the ocean at this place (when *rg_opt_weather_runoff=.true.*). Depending on the primary source rock, we tied the rate $M$ of Sr, Os or Li ion delivery to the ocean to Ca and magnesium (Mg) input rates from weathering of carbonates ($Ca_{CaCO_3}$) and/or silicates ($Ca_{CaSiO_3}$), related by a constant ratio $k$:

$$M_{CaCO_3} = k_{CaCO_3} \cdot Ca_{CaCO_3} \tag{1}$$

$$M_{CaSiO_3} = k_{CaSiO_3} \cdot Ca_{CaSiO_3} \tag{2}$$

ROKGEM allows for different $CaSiO_3$ weathering schemes, including one which separates total $CaSiO_3$ weathering into contributions from granite and basalt weathering. For this case, we imposed individual parameters for Sr weathering for each lithology. The amount and isotopic composition of weathering-derived Os is mostly dependent on organic matter content (Jaffe et al., 2002; Georg et al., 2013; Dubin and Peucker-Ehrenbrink, 2015), but the 0D weathering scheme in ROKGEM only differentiates between Os delivery from $CaSiO_3$ and $CaCO_3$ weathering. To enable a more realistic simulation of continental Os weathering fluxes, we extended the 2D weathering schemes in ROKGEM to trace Os and its isotopes. While computationally more expensive, Os weathering fluxes can be tied to specific lithologies in these schemes, including organic-rich shales.

The effect of secondary mineral formation on land on Li concentrations and isotope abundances in continental run-off can be simulated based on the empirical relationships between the weathering Li flux ($Li_{CaSiO_3}$) and the ratio of weathering to denudation rate ($WD$, also referred to as weathering congruency) reported in Pogge von Strandmann et al. (2020):

$$Li_{CaSiO_3} = k_{CaSiO_3} \cdot Ca_{CaSiO_3} \cdot e^{0.4883 \cdot log(WD)} \tag{3}$$

cGENIE derives an estimate of $WD$ from the weathering modifier which adjusts the chemical weathering flux from silicates on the basis of a deviation of climate (e.g. surface land temperature) from some reference value (and a value of $WD$ of 1.0). Mantle-derived metal input via hydrothermal vents is parameterised in the SEDGEM module.

To represent the vast range of chemical conditions and reaction rates at the sediment-water interface in the ocean efficiently, three biogeochemically distinct depositional environments are represented in SEDGEM: (1) dynamic shallow seas with reef-building biota ('reef'), (2) organic rich sediments depleted in oxygen ('muds'), and (3) plankton-derived carbonate deposition

in the open ocean ('deep sea'). CaCO$_3$ deposition and dissolution as well as elemental fluxes across the sediment-water interface are parameterised differently in each of these environments, with masks used to distinguish the different environments. Hydrothermal inputs of elements into bottom waters occurs only in grid cells which represent 'deep sea'. We parameterised this flux such that a global total input flux is prescribed, which can either be distributed equally across 'deep sea' grid cells or according to an easily adjustable, spatially-explicit map (e.g. following the line of mid ocean ridges). There is no differentiation between seafloor areas with predominant high- or low-temperature hydrothermal venting in our implementation. Hence, we suggest prescribing net hydrothermal metal fluxes with an averaged isotopic composition in cGENIE to account for metal inputs under both temperature regimes.

Recrystallization of SrCO$_3$ is an additional source of Sr to the water column (see above). Its parameterisation is similar to that of hydrothermal inputs, except that elemental fluxes from recrystallization only occur in bottom waters of grid cells labelled as 'reef'. The size and rate of this Sr flux can either be set as a total global or an area-weighted value.

To account for inputs at the air-sea interface (only assumed to be relevant for Os, as described in the previous section), we added the option to prescribe a spatially explicit field of annual input into the surface ocean (or technically anywhere into the water column).

## 3.2 Marine Sr, Os, Li and Ca sinks

Ca export from the surface ocean is linked to the export of particulate organic carbon (POC) by a constant CaCO$_3$:POC ratio (rain ratio) (see Ridgwell et al., 2007, for a description of POC export simulation in cGENIE). We parameterise the incorporation of Os, Li and Sr into carbonates by scaling their export in carbonates to the Ca export. This can be done by setting either a constant Sr/Li/Os-to-Ca ratio ($r_{M/Ca}$) or a constant scaling factor $\alpha$ which links the local Sr/Li/Os-to-Ca ratio in seawater to the Sr/Li/Os-to-Ca ratio in the precipitate (following e.g. Tang et al., 2008):

$$export_{metal} = k \cdot export_{Ca} \tag{4}$$

$$k = r_{M/Ca} + \alpha \cdot [Metal]_{water}/[Ca]_{water} \tag{5}$$

The Sr/Li/Os-to-Ca ratio of the precipitate is then used to calculate the metal consumption during carbonate formation, as well as metal release during carbonate dissolution. To reflect the different Sr/Li/Os-to-Ca ratios in pelagic compared to reef carbonates, a separate factor can be set for reef production. Reef CaCO$_3$ immediately contributes to surface sediments, while pelagically-formed CaCO$_3$ sinks through the water-column, and might be dissolved before reaching the 'deep sea' sediment-water interface as a result of local chemical equilibria. cGENIE also calculates Li export by abiotically precipitated carbonates.

We further included the sequestration of Sr and Li in seafloor sediments due to seafloor weathering by scaling the metal flux into the uppermost sediment layer ($fMetal$) to the metal concentration of the overlying ocean grid cell ($[Metal]$):

$$fMetal = k \cdot [Metal] \tag{6}$$

This burial occurs in all grid cells that are labelled as 'deep sea'. In the absence of better constraints on depositional mechanisms, we include a mathematically similar sink for Os. For a desired global burial rate $B$ (mol s$^{-1}$), the value of $k$ can be estimated by considering the total area of 'deep sea' $A_{deep}$ (m$^2$) and equilibrium metal concentration $[Metal]$ (mol kg$^{-1}$):

$$k = \frac{B/A_{deep}}{[Metal]} \tag{7}$$

5    Our model calculates Li burial during clay formation locally at the sea-sediment interface. The burial flux $fLi_{clay}$ is scaled to the concentrations of Li in the deepest grid box of the water column and the detrital flux into the sediments:

$$fLi_{clay} = k \cdot [det] \cdot [Li] \tag{8}$$

To test implications of Os burial with particulate organic carbon, we included an option for [O$_2$]-sensitive scavenging from the water column. In the surface ocean, organic carbon is exported as a function of the concentration of inorganic carbon, light, nutrient availability (PO$_4^{3-}$ in our set up), sea ice cover and temperature (Ridgwell, 2001). This export production of organic carbon is split into dissolved and particulate organic carbon (DOC and POC) at an adjustable ratio. DOC is advected and diffused in the ocean, while POC is instantly exported to deeper water layers. Remineralization of organic matter can either be set to follow empirical decay functions, or to depend on local temperature and redox conditions. Depending on which chemical species are turned on for an experiment, the redox state is calculated based on oxygen, sulfate, methane, nitrate and iron oxide concentrations. O$_2$ is exchanged between the atmosphere and the ocean at the air-sea interface, depending on the solubility of O$_2$ in water. In the ocean, O$_2$ is released during primary production in the surface ocean and consumed during remineralization of DOC and POC under oxic conditions. Sulphur, iron, methane and nitrogen cycling are less relevant for the redox state in the modern open ocean but influenced POC fluxes in the past, and are described in detail in van de Velde et al. (2020), Reinhard et al. (2020) and Naafs et al. (2019). Following the example of existing scavenging parameterisation schemes in cGENIE, we implemented this process by scaling the flux of scavenged Os ($f_{scav}$) to the local concentrations of Os and $POC$:

$$f_{scav} = k \cdot [Os] \cdot [POC] \tag{9}$$

Since there is evidence that Os needs to be reduced in order to be buried, we include a switch to use the scavenging code only where [O$_2$] is below an adjustable threshold.

### 3.3  Isotopes

25  cGENIE tracks isotopes in total molar abundances rather than delta notation or ratios so that they can be advected and diffused like any other tracer. Li has two stable isotopes, and thus its implementation follows that of other elements with two stable isotopes (e.g. C, Ridgwell (2001)). The addition of Sr, Os and Ca isotopes needed different approaches because they have more than two principal stable isotopes. We chose to reduce the number of traced isotopes to three for Sr and Os and two for Ca, because these subsets contain the most abundant isotopes of the respective element (Sr) and/or are most relevant for their

application as seawater proxies (Sr, Os and Ca). Calcium is thus currently treated like an element with two stable isotopes in cGENIE. To track three isotopes rather than two, we track two isotopes explicitly and one implicitly with the bulk abundance of the trace metal. In the case of Sr, abundances of $^{87}$Sr and $^{88}$Sr are tracked explicitly. The abundance of $^{86}$Sr is then taken as the difference between the abundances of the two explicitly tracked isotopes and the bulk Sr abundance, since $^{84}$Sr can be neglected. Os has more than two stable isotopes, but only two of them, $^{187}$Os and $^{188}$Os, are currently used as a proxy system and so these two isotopes are tracked explicitly and the bulk Os abundance contains all remaining stable isotopes.

Every metal flux in the model is accompanied by fluxes of the traced isotopes. The scale of these isotope fluxes depends on the size of the metal flux and the relative abundance of the respective isotope, derived from the model configuration in which the model user prescribes the isotopic composition of marine metal in- and outputs. For simplicity, the user can prescribe these isotopic compositions in delta notation for stable isotopes and ratios for radiogenic isotopes. The model then converts these values into molar isotope abundances, using the hard-coded isotope standards listed in table 5 (international standards where available and observations of modern-day sea water otherwise).

**Table 5.** Isotopic standards in cGENIE.

| Isotope ratio | Standard composition | Standard material/reference |
|---|---|---|
| $^{87}$Sr/$^{86}$Sr | 0.709175 | Mokadem et al. (2015) |
| $^{88}$Sr/$^{86}$Sr | 8.375209 | NBS987 |
| $^{187}$Os/$^{188}$Os | 1.05 | Lu et al. (2017) |
| $^{188}$Os/$^{189+190+192}$Os | 0.159 | Dąbek and Halas (2007) |
| $^{7}$Li/$^{6}$Li | 12.33333 | L-SVEC |
| $^{44}$Ca/$^{40}$Ca | 0.021229 | NIST SRM915a Heuser et al. (2002) |

Given the current lack of evidence for Os isotopic fractionation outside of the lithosphere (e.g. Nanne et al., 2017) we do not include a fractionation factor for marine Os sinks. For Sr and Li, we include isotopic fractionation during carbonate and secondary mineral formation. In its default setting, cGENIE uses constant fractionation factors for Sr, Li and Ca fluxes, but we included optional schemes to simulate environmental controls on stable Li and Ca isotope fractionation. For Li, we included optional corrections of the riverine $\delta^7$Li for weathering congruency (following Pogge von Strandmann et al. (2020)) and the temperature sensitivity of Li isotope fractionation during terrestrial and marine clay formation (Millot et al., 2010):

$$\delta^7 Li_{runoff} = \delta^7 Li_{CaSiO_3} - 5.4079 \cdot log(WD) \tag{10}$$

$$\delta^7 Li_{runoff} = \delta^7 Li_{runoff} + k \cdot (T - T_0) \tag{11}$$

$$\Delta^7 Li_{burial} = \Delta^7 Li_{burial} + k \cdot (T - T_0) \tag{12}$$

with $\delta^7\text{Li}_{CaSiO_3}$ being the assumed $\delta^7\text{Li}$ of weathered silicates at the reference temperature, $k$ the isotopic effect of a $1°\text{C}$ temperature change and $T_0$ being the reference temperature. A consistency check prevents the resulting $\delta^7Li_{runoff}$ from falling below the composition of continental crust. If these functions are not used, the absence of simulated Li fractionation in freshwater should be taken into account by setting the parameters for terrestrial Li input based on the composition of river water rather than weathered Li at the reference temperature. cGENIE also offers two different choices for Ca isotope fractionation during carbonate formation (in addition to the default of no fractionation), carbonate ion concentration-dependent fractionation following Gussone et al. (2005) and saturation state ($\Omega$)-dependent fractionation following Tang et al. (2008)):

$$\Delta^{44/40}Ca_{CaCO3} = -1.31 + 3.69 \cdot [CO_3^{2-}] \cdot 10^3 \qquad \Delta^{44/40}Ca_{CaCO3} = -0.066649 \cdot \Omega - 0.320614 \qquad (13)$$

These fractionation schemes are explored in the supplementary material of this manuscript (SI section C) as well as in Fantle et al. (2020).

The model parameters required to set isotopic fractionation of Os, Li and Sr are listed in table A8.

## 4   Model configuration and validation

Box models are important tools to investigate the global balance of elemental and isotopic fluxes and essential to estimate elemental residence times as well as to identify missing sinks and sources (e.g. Stoffyn-Egli and Mackenzie, 1984). They are usually validated with observed average seawater compositions (e.g. Krabbenhöft et al., 2010; Misra and Froelich, 2012; Pearce et al., 2015). The three-dimensional grid of cGENIE allows for comparison of the spatial pattern produced by the simulated set of processes to observations, which can further bolster or challenge our assumptions about global trace metal cycling on diverse time scales. Here we set up the model to represent the pre-industrial state of trace metal cycling, and compare the model output to measured modern seawater profiles.

### 4.1   Pre-industrial configuration

We initialised cGENIE with a modern-day geography on a 36x36 grid with up to 8 depth layers in the ocean (following Ridgwell and Hargreaves, 2007) and with a pre-industrial $CO_2$ concentration of 278 ppm. Marine biological productivity is simulated using a single nutrient scheme as outlined in Ridgwell et al. (2007) but here with a constant $CaCO_3$:POC rain ratio set to 0.043 in-line with typical paleo configurations of the model (Panchuk et al., 2008). Burial and dissolution of $CaCO_3$ in deep-sea sediments follows Ridgwell and Hargreaves (2007). This configuration results in a pelagic $CaCO_3$ burial rate of 0.125 PgC yr$^{-1}$, close to the observationally-calibrated $CaCO_3$ sink of Ridgwell and Hargreaves (2007). In addition to the pelagic environment, covering 350.6 million km$^2$, we simulated reef deposition on shelves, which covers 5.5 million km$^2$ in our particular low-resolution modern model grid (see SI D, the sediment model bathymetry was derived from ETOPO5 (Data Announcement 88-MGG-02, 1988)). Reefal deposition was simulated in grid cells representing marine sediments shallower than 1000 m and not further pole-wards than $41.8°$ N/S, on the basis that reefal carbonate deposition is predominantly a tropical

and subtropical process. We prescribe a reefal $CaCO_3$ sink of 0.05 PgC yr$^{-1}$ associated with these environments. Temperature dependent terrestrial silicate and carbonate weathering (Lord et al., 2016) were included with the baseline temperature and rates of carbonate and silicate weathering set to 8.48°C, 8.4 Tmol yr$^{-1}$ and 6.0 Tmol yr$^{-1}$, respectively. Silicate weathering was split into weathering of $CaSiO_3$ (2/3) and $MgSiO_3$ (1/3). In the absence of a numerical representation of hydrothermal cation exchange (Coogan and Gillis, 2018), we balanced the Ca and Mg cycles by including a constant exchange between sedimentary Ca and dissolved Mg of 2.0 Tmol yr$^{-1}$ at the seafloor. To close the long-term carbon cycle, we prescribed a fixed rate of organic C burial of 0.031 PgC yr$^{-1}$ with $\delta^{13}$C = -30 ‰ as well as a total exogenic carbon influx of 0.103 PgC yr$^{-1}$ with $\delta^{13}$C = -6 ‰. This net input can be regarded as the result of 0.103 PgC yr$^{-1}$ subaerial outgassing with $\delta^{13}$C = -4.6 ‰ (Mason et al., 2017), hydrothermal emissions of 0.018 PgC yr$^{-1}$ with $\delta^{13}$C = -6 ‰ at mid-ocean ridges and consumption of 0.018 PgC yr$^{-1}$ with $\delta^{13}$C = 2 ‰ during seafloor weathering (Cocker et al., 1982).

Table 6 summarizes the trace metal fluxes we prescribed to simulate the pre-industrial trace metal cycling, consistent with the observational constraints provided in tables 1 - 4. We used the 0D weathering scheme for computational efficiency, but since a differentiation between carbonate and silicate derived Os does not capture the lithology-dependence of the Os weathering flux appropriately, we prescribed the net abundance and isotopic composition of Os in continental run-off rather than lithology-specific compositions. We also did not simulate Os scavenging in these pre-industrial spin-ups because the dependence of marine Os burial on organic matter abundance and redox state are uncertain. However, in Appendix A1 and A2 we included the results of two pre-industrial Os cycle spin-ups – one with the 2D weathering scheme and the second with Os scavenging under occurring anoxic conditions and accounting for 50% of marine Os burial. Note that the prescribed hydrothermal metal fluxes are net fluxes of high- and low-temperature hydrothermal activity. The specific model parameters that need to be prescribed to achieve these fluxes are given in the SI of this manuscript and in the github repository containing all relevant configuration files, which can be accessed as outlined under 'code availability'.

By applying the chosen model parameters for our spin-up with modern boundary conditions and running the model to steady state, we assumed that metal fluxes are currently in equilibrium. This is a common assumption when modelling weathering tracer isotopes and their perturbations in the geologic record (e.g. Misra and Froelich, 2012; Bauer et al., 2017). However, given the long residence time estimates in today's ocean, it is likely that at least the Sr, Li and Ca cycles are not fully in steady state today (i.e. Derry, 2009). For example, Paytan et al. (2021) inferred that the marine Sr has constantly fluctuated over the last 35 Myr. In contrast, constant Os isotopic compositions of seawater over the last millennia suggest that Os has reached steady state since the last deglaciation, but residence time estimates of >20 kyr based on modern day Os fluxes put this into question (Oxburgh, 2001). The newly implemented tracers in cGENIE can be used to study different equilibria and transient adaptations of the marine metal reservoirs to new boundary conditions. For example, we could not simulate the observed marine Sr reservoir with the observed Sr input, which indicates an imbalance observed Sr fluxes and the marine Sr reservoir. Hence, we simulated two different steady states for Sr under pre-industrial boundary conditions: One with a best estimate of pre-industrial Sr fluxes (hereafter referred to as FLUXES) and one tuned to best fit the spatial mean of observations (TUNED).

**Table 6.** Metal fluxes for pre-industrial spin up.

| Process | | Flux (mol yr$^{-1}$) | Isotopic composition | |
|---|---|---|---|---|
| **Strontium** | | | | |
| | | | $^{87}$Sr/$^{86}$Sr | $\delta^{88/86}$Sr (‰) |
| weathering | FLUXES | 34.0x10$^9$ | 0.7122 | 0.318 |
| | TUNED | 31.0x10$^9$ | 0.7097 | 0.256 |
| hydrothermal input | | 3.1x10$^9$ | 0.7038 | 0.261 |
| diagenetic input | FLUXES | 3.4x10$^9$ | 0.708 | 0.385 |
| | TUNED | 3.4x10$^9$ | 0.7087 | 0.256 |
| carbonate sink | FLUXES | 40.5x10$^9$ | seawater | seawater - 0.18 |
| | TUNED | 37.5x10$^9$ | seawater | seawater - 0.18 |
| **Osmium** | | | | |
| | | | $^{187}$Os/$^{188}$Os | $^{188}$Os/$^{189+190+192}$Os |
| weathering | | 2605.26 | 1.2 | 0.159 |
| hydrothermal input | | 526.32 | 0.5625 | 0.159 |
| aeolian input | | 157.89 | 0.2 | 0.159 |
| sediment deposition | | 3289.47 | seawater | seawater |
| carbonate sink | | 0.0 | seawater | seawater |
| **Lithium** | | | | |
| | | | $\delta^7$Li (‰) | |
| weathering | | 8x10$^9$ | 23.0 | |
| hydrothermal input | | 6x10$^9$ | 8.3 | |
| secondary mineral formation sink | | 4x10$^9$ | seawater -15.0 | |
| sea floor alteration sink | | 9.5x10$^9$ | seawater -15.0 | |
| carbonate sink | | 0.5x10$^9$ | seawater - 4 | |
| **Calcium** | | | | |
| | | | $\delta^{44}$Ca (‰) | |
| weathering | | 12.4x10$^{12}$ | 0.9 | |
| hydrothermal input | | 2x10$^{12}$ | 0.93 | |
| carbonate sink | | 14.4x10$^{12}$ | seawater - 1.1 | |

The model spin up was carried out in three stages to improve computational efficiency. The first stage (20 kyr) was run with a closed marine carbonate system, where the marine C and alkalinity reservoirs are artificially restored by balancing losses through CaCO$_3$ burial with external inputs to the ocean. During this stage, the climate and ocean dynamics adjust

to the physical boundary conditions and ocean-atmosphere C exchanges balance. In the second phase (500 kyr) the marine carbonate system was open so that prescribed inputs from terrestrial weathering and the mantle and marine burial dynamically adjusted to balance one another. During the third stage (15 Myr), when ocean dynamics, C, nutrient and Ca cycles were already equilibrated, the prescribed Sr, Os and Li sources were added and the model was run until metal concentrations and

their isotopic composition in the ocean were at steady state. The model calculations were accelerated during the last two stages of the spin up using the time-stepping method introduced by Lord et al. (2016).

## 4.2  Comparison between simulated and observed trace metal contents of seawater

One advantage of simulating trace metal cycles within a 3D earth system model is that we can test, for the first time, simulated metal concentrations and isotopes against observed spatial patterns. In the following sections, the simulated pre-industrial

distribution of each metal is compared against observational data shown in section 2.5.

### 4.2.1  Strontium

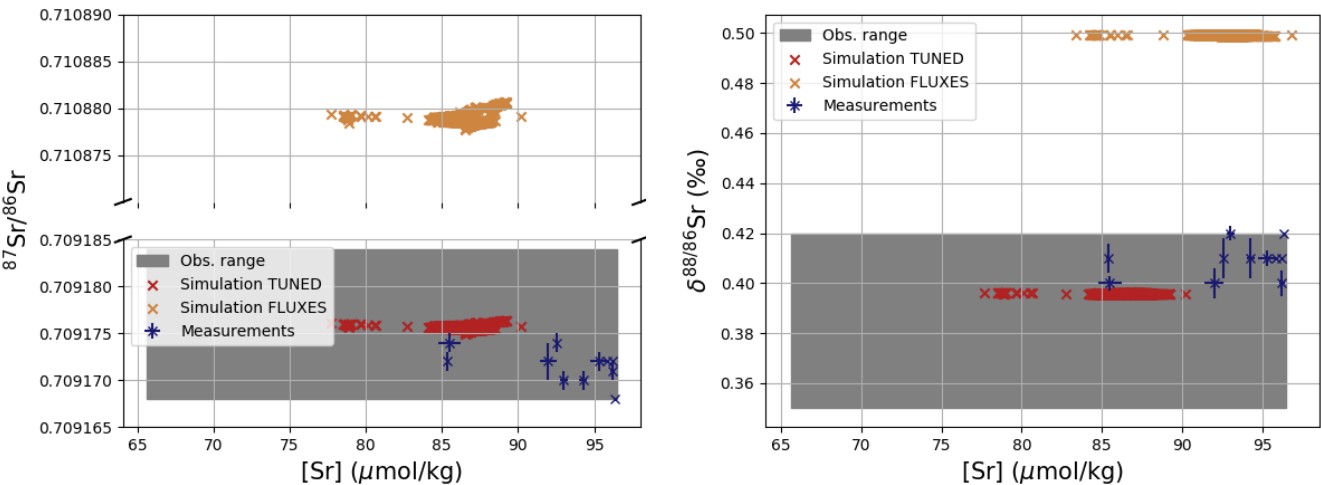

**Figure 3.** Comparison between measured and simulated concentrations and isotopic composition of dissolved Sr in seawater. The displayed data points are paired measurements of Sr concentrations and isotope ratios (Wakaki et al., 2017), while the gray box also indicates the full range of concentrations and isotopic compositions (separate measurements of Sr concentration or isotope ratios) (Angino et al., 1966; Fabricand et al., 1967; Bernat et al., 1972; Brass and Turekian, 1974; De Villiers, 1999; Mokadem et al., 2015). Observations and model results are salinity-normalized.

Figure 3 compares Sr concentrations and isotopic compositions in our simulation with observations. The simulated marine Sr reservoir is more homogeneous than the observed in terms of concentrations and isotopic composition. The FLUXES simulation, in which we prescribe observationally-constrained marine Sr inputs, results in more radiogenic and isotopically heavy

marine Sr than observed. To yield a steady state marine $^{87}$Sr/$^{86}$Sr close to observations, we need to assume that continental Sr

supply is on average less radiogenic than observed (in agreement with e.g. Pearce et al., 2015). Similarly, the observed fractionation factor for stable Sr isotopes during biogenic $CaCO_3$ formation is too big to equilibrate the model at the observed low $\delta^{88/86}Sr$ if we do not assume that all Sr sources only provide mantle-like light Sr ($\delta^{88/86}Sr = 0.256‰$, TUNED simulation). Corroborating the conclusion of e.g. Vance et al. (2009), this suggests that the isotopic composition of marine dissolved Sr is
5  currently not at equilibrium.

The simulations capture the homogeneous vertical profiles of Sr concentrations observed at most sites (see figures B1 and B5 in the SI). One exception are modelled Sr concentrations in the North Atlantic which are high compared with older measurements. Given that these low observed North Atlantic values come from one of the first studies measuring Sr concentrations in seawater (Angino et al., 1966), and that other studies of North Atlantic seawater yielded higher Sr concentrations (e.g. De Vil-
10  liers, 1999), this difference could be partially the result of analytical errors, although influences from seasonal/inter-annual variability not captured by the model cannot be ruled out.

### 4.2.2  Osmium

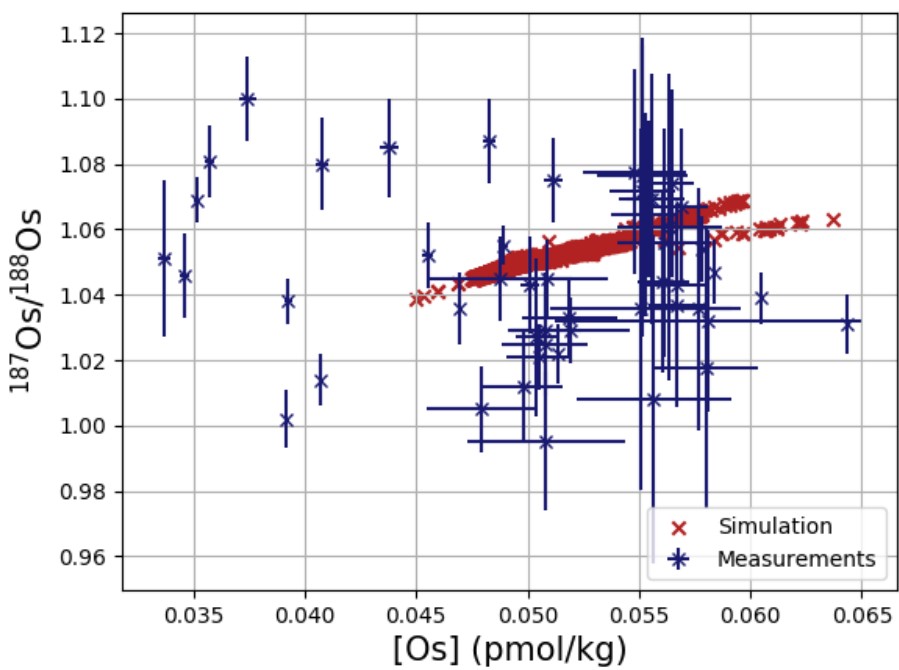

**Figure 4.** Comparison between measured and simulated concentrations and isotopic composition of dissolved Os in seawater. Shown are all available measured Os concentrations and isotope ratios for the modern ocean alongside the values of all ocean grid cells in the simulation. Data is taken from Levasseur et al. (1998), Woodhouse et al. (1999) and Gannoun and Burton (2014).

Os concentrations and isotopes are generally more spatially homogeneous in the simulation compared with observations. For 80% of the paired measurements of Os concentrations and isotopic compositions, one or more cGENIE simulated grid cells fall within measurement uncertainty. The remaining 20% of the Os observational data (those measurements with the lowest Os concentrations and isotopic ratios) were not reproduced in any grid cells in our simulation. This discrepancy between model

simulation and observations is not the result of the simplified homogeneous terrestrial Os input in the model. In a repetition of the model spin up with spatially variable Os input (see SI A1), we found no appreciable change to the simulated ranges of Os concentrations and isotope ratios in the oceans, i.e. no appreciable increase in spatial heterogeneity. Since the low Os concentrations not captured by the model are observations from surface and intermediate waters from the East Pacific (fig. B6, Woodhouse et al., 1999; Gannoun and Burton, 2014), some processes affecting Os removal in that region might instead be

missing in our model. For example, Woodhouse et al. (1999) suggested that Os could bind onto organic matter in the water column under low oxygen conditions. We tested the effect of increased Os removal with particulate organic matter from anoxic water masses in cGENIE by associating half of the marine Os burial with the optional Os scavenging sink (equation 9, see the results of our sensitivity study in SI A2). The true magnitude of Os burial associated with organic matter is uncertain but we assumed here that it is responsible for all suboxic Os removal, which is estimated to account for at least the 50% of total marine

burial flux we assumed (Lu et al., 2017). Adding such dependence of Os burial on environmental conditions increased the range of simulated Os concentrations beyond that observed, i.e. Os scavenging is more than capable of producing the observed spatial heterogeneity in Os concentrations. However there is no known Os isotopic fractionation associated with scavenging or burial; in the absence such fractionation, scavenging does not increase the range of simulated marine Os isotope ratios. Observations from sites outside the East Pacific show no vertical Os concentration gradients, which is well-reproduced by our simulation set-

up. The Os isotopic composition shows a uniform vertical distribution in the simulations, which is consistent with observations at all sites (see figures B6b). While our initial simulation is thus consistent with the majority of Os measurements, we showed that cGENIE provides the necessary framework to investigate regionally-variable environmental controls on the marine Os reservoir.

### 4.2.3 Lithium

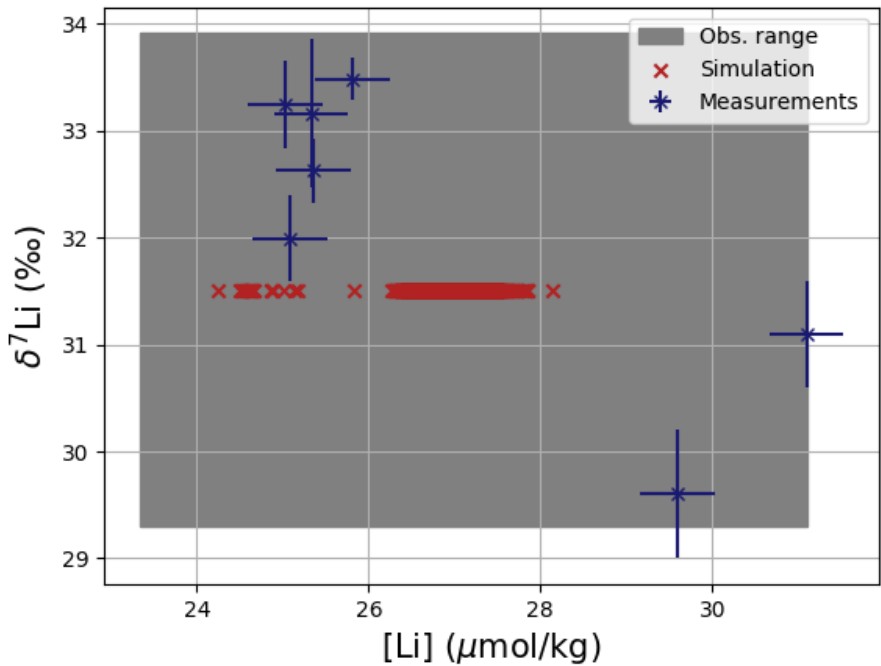

**Figure 5.** Comparison between measured and simulated concentrations and isotopic composition of dissolved Li in seawater. The displayed data points are measurements of Sr concentrations and isotope ratios on the same samples (Hall, 2002), while the gray box also indicates the range of concentrations and isotopic compositions that were measured separately (Angino and Billings, 1966; Fabricand et al., 1967; Chan, 1987; Chan and Edmond, 1988; You and Chan, 1996; Moriguti and Nakamura, 1998; Tomascak et al., 1999; James and Palmer, 2000; Košler et al., 2001; Nishio and Nakai, 2002; Bryant et al., 2003; Pistiner and Henderson, 2003; Millot et al., 2004; Choi et al., 2010; Pogge von Strandmann et al., 2010; Lin et al., 2016; Henchiri et al., 2016; Phan et al., 2016; Weynell et al., 2017; Fries et al., 2019; Gou et al., 2019; Hindshaw et al., 2019; Murphy et al., 2019; Pogge von Strandmann et al., 2019). Observations and model results are salinity-normalized.

Similar to Sr and Os, the model simulations fall within the range of observed concentrations and isotopic compositions of dissolved Li, although they are more homogeneous than observed (see fig. 5). cGENIE can be used to explore whether environmental influences on local Li inputs and burial can explain this disparity, though under-estimated measurement uncertainties and inter-laboratory inconsistencies, particularly in older studies, could also contribute to the spread in observed values. For example, the high concentrations which are underrepresented in our simulation (see fig. B7 and fig. B3) come from one study in the Caribbean Sea (Angino and Billings, 1966), a marginal ocean which is not representative of the open ocean and which might not be sufficiently resolved in the coarse grid of cGENIE. Yet, Angino and Billings (1966) was one of the first studies measuring Li concentrations in seawater, and their measurements may be associated with much larger uncertainties than

reported. Were these measurements excluded, the spatial mean of observed Li concentrations would be lower, and the model could be re-tuned by prescribing slightly lower Li fluxes in- and out of the ocean which are permissible given the current range of flux estimates (see table 3).

Only one measured vertical profile of Li isotopes is available in the literature (Hall, 2002), and it shows no vertical $\delta^7$Li variation, which is reproduced by our simulation. The simulated $^7$Li is offset from the profile reported by Hall (2002) since we tuned the model to the average of reported seawater $^7$Li measurements but the profile in Hall (2002) contains some of the most $^7$Li enriched published seawater values.

### 4.2.4  Calcium

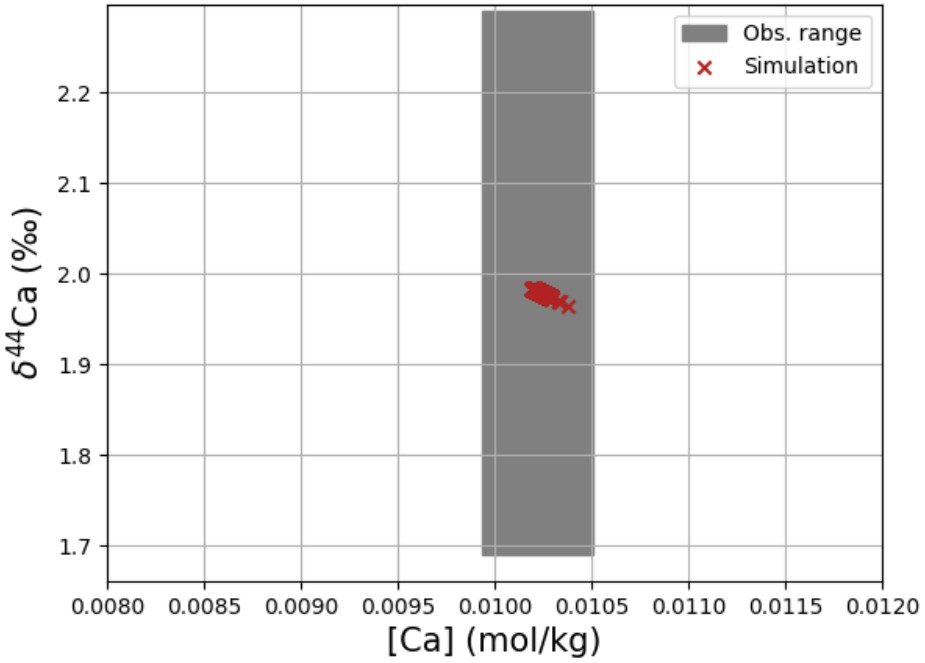

**Figure 6.** Comparison between measured and simulated concentrations and isotopic composition of dissolved Ca in seawater. The gray box indicates the range of concentrations and isotopic compositions that were measured on separate samples (Fabricand et al., 1967; De Villiers, 1999; Fantle and Tipper, 2014). Observations and model results are salinity-normalized.

Simulated concentrations of dissolved Ca and its homogeneous isotopic composition closely resemble the observations, the spread in observed Ca isotope ratios likely being due to measurement uncertainty given the large reported error bars (see fig. 1). Repeating our model spin up with different parameterizations of Ca isotope fractionation during biogenic carbonate formation did not appreciably increase the simulated range of marine Ca isotope ratios (see our sensitivity study in SI C). Similar to the

average Ca concentration in seawater, the uniform vertical distribution is well captured by the model simulation (see fig. B8). Likewise, no vertical Sr/Ca gradients are observed or simulated. Observed Sr/Ca ratios are in between simulation results with tuned and un-tuned marine Sr reservoir. This is an artefact of the tuning to mean observed Sr concentrations which is lower than the mean Sr concentrations of the sites shown in figure B8 because it includes the anomalously-depleted samples from the North Atlantic. This mismatch is another indication that the Sr measurements reported in Angino et al. (1966) might be associated with a larger uncertainty than reported.

## 4.3 Transient perturbation experiments

Another advantage of the new metal cycles in cGENIE is that we can study their transient behaviour under external forcings in a fully coupled system, including effects from ocean circulation and primary productivity changes, climate-sensitive weathering and marine carbonate accumulation and dissolution. Here we present one such example, where we instantaneously release either 1000 Pg C or 5000 Pg C into the atmosphere to produce temporary weathering responses which re-equilibrate the model (similar to Lord et al., 2016). These scenarios allow us to showcase the sensitivity of the simulated metal systems and to discuss the Earth system processes that shape their transient evolution. Our experiments are set up to understand model behaviour rather than to compare against any particular geological records or represent any particular event, although the choices of carbon release relate to estimates associated with differing 'hyperthermal' events of the Paleogene (e.g. Panchuk et al., 2008; Kirtland Turner and Ridgwell, 2013). It is also important to note that some of these scenarios generate isotopic perturbations smaller than the analytical uncertainty of real-world measurements.

**Table 7.** Marine trace metal concentrations and residence times in our pre-industrial spin up compared to those reported in literature (see scientific background).

| Metal | Mean concentration (mol kg$^{-1}$) | Simulated residence time (yr) | Residence time reported in literature (yr) |
|---|---|---|---|
| Strontium FLUXES | $92.84 \times 10^{-6}$ | ~2,100,000 | 1,900,000–3,450,000 |
| Strontium TUNED | $86.50 \times 10^{-6}$ | ~1,960,000 | 1,900,000–3,450,000 |
| Osmium | $52.64 \times 10^{-15}$ | ~21,500 | 3,000–50,000 |
| Lithium | $27.00 \times 10^{-6}$ | ~2,600,000 | 300,000–3,000,000 |
| Calcium | $10.25 \times 10^{-3}$ | ~960,000 | 500,000–1,300,000 |

In our spin-up, the prescribed metal fluxes result in residence times within the range of published estimates (table 7). Li has the longest residence time in our simulations, and thus the largest inertia to external perturbations. Sr and Ca follow, with Os being the most responsive system.

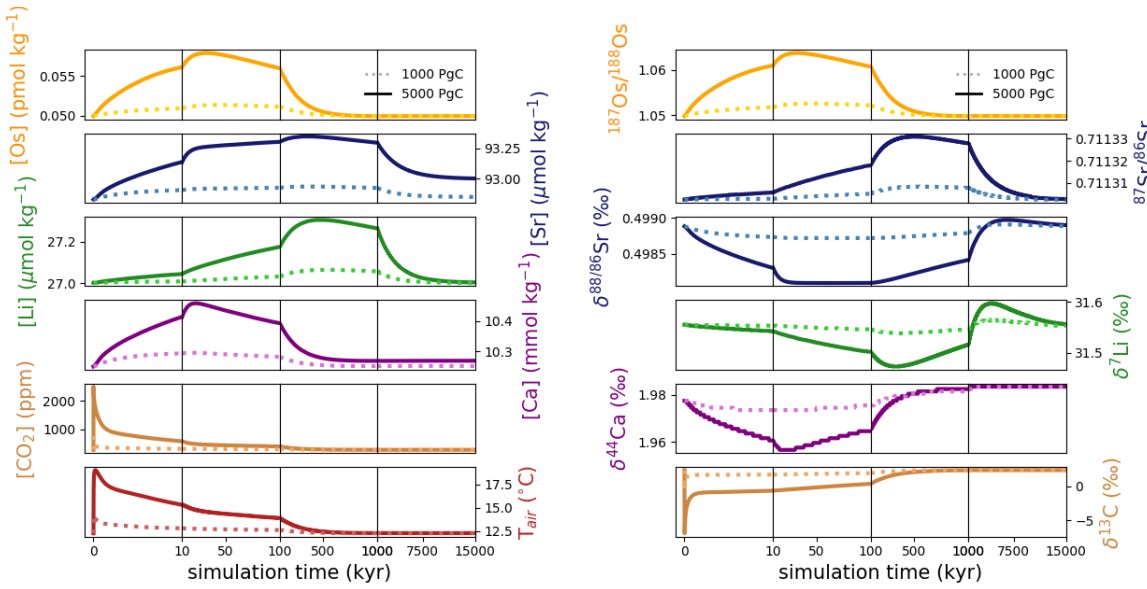

**Figure 7.** Transient changes in modelled Sr, Os, Li, Ca, composition of seawater and atmospheric $CO_2$ and temperature following instant releases of 1000 Pg C and 5000 Pg C. Shown are changes a) in concentrations, and b) in isotopic composition of seawater.

Fig. 7 shows the modelled transient responses of Sr, Os, Li, Ca concentrations and isotopic compositions, DIC $\delta^{13}C$, atmospheric $CO_2$ and temperature to an exogenic $CO_2$ pulse. After the instantaneous carbon release, atmospheric $CO_2$ concentration increases and the $\delta^{13}C$ of seawater decreases on the timescale of years. The sudden increase in atmospheric $CO_2$ concentrations replenishes the marine carbon reservoir, which decreases biogenic carbonate formation in the surface ocean and leads to the dissolution of pelagic carbonates. Over the next ∼10,000 yrs, this negative carbon cycle feedback removes the majority of the initial excess carbon from the atmosphere (Lord et al., 2016). Increased atmospheric $CO_2$ also enhances the greenhouse effect and thus leads to a rise in global mean air temperature, altering the hydrological cycle and speeding up reaction rates which result in increased rates of continental crust weathering. After a few tens of thousands of years, increased continental weathering supplies enough alkalinity to the ocean to slow down carbonate dissolution and increase carbonate burial. Through enhanced silicate weathering and carbonate burial, the remaining excess carbon is sequestered over the timescale of ∼100,000 yrs after the initial emissions pulse. Over the same timescale, temperature, which due to Earth system feedbacks initially recovers more slowly than atmospheric $CO_2$ concentrations, returns to pre-emissions values and ends the phase of enhanced weathering.

Metal input from continental weathering (and marine carbonate dissolution for Sr and Ca) leads to a transient growth of marine metal reservoirs and isotopic excursions. Increased continental weathering increases the influx of radiogenic Sr and Os, driving the respective marine reservoirs to more radiogenic isotopic compositions. Continental crust is also depleted in heavy

stable isotopes compared to seawater, and the increased weathering influx therefore reduces the $\delta^{88/86}$Sr, $\delta^7$Li and $\delta^{44}$Ca values of seawater. The isotopic composition of carbonates and silicates in continental crust is spatially uniform in our set-up and cGENIE does not simulate any isotopic effects associated with changes in erosion rate. By default, isotopic shifts due to changes in secondary mineral formation and dissolution in freshwater systems also are not simulated. Yet, just such a shift has

5 been deduced as enhancing the magnitude of $\delta^7$Li isotope excursions observed in the geologic record (e.g. for $\delta^7$Li, Lechler et al., 2015). Indeed, enabling the climate-sensitive parameterization of $\delta^7$Li in continental run-off demonstrates that this can increase the excursion amplitude to be more consistent with perturbations in the geologic record (see our sensitivity study in SI B). The isotopic composition of weathering-derived inputs of Sr, Os and Ca, which have different isotopic compositions in carbonate and silicate rocks, also varies in our simulation because of the different pace with which the weathering of these

10 lithologies changes under climate change. For example, the radiogenic Sr excursion in the ocean is amplified by a transient increase in $^{87}$Sr/$^{86}$Sr of weathering-derived Sr.

Our model is set up so that alkalinity and $Ca^{2+}$ inputs to the ocean are equal in steady state. Since the alkalinity supply is entirely derived from climate-sensitive continental weathering but 1/3 of the $Ca^{2+}$ influx is derived from a constant hydrothermal source, the climate-driven increase in weathering results in a transiently larger alkalinity than $Ca^{2+}$ supply which

does not appear in C cycle simulations without an igneous $Ca^{2+}$ source at the seafloor (e.g. Lord et al., 2016; Vervoort et al., 2019). The imbalance is small (10% of the Ca input at its peak) but cumulatively the excess alkalinity supply is large enough to cause a slight reduction of the steady-state marine Ca and Sr reservoirs (by $\sim$0.5% following the emission of 5000 Pg C). Our assumption of a climate-insensitive $Ca^{2+}$ source at the seafloor is a simplification since there is evidence that this flux varies with bottom water properties, in particular temperature (e.g. Krissansen-Totton and Catling, 2017). However, the tem-

perature sensitivity of the seafloor $Ca^{2+}$ influx is likely lower than that of continental weathering (Brady and Gíslason, 1997) and temperature change is dampened in the deep ocean. Thus, even if the temperature sensitivity of hydrothermal $Ca^{2+}$ input was considered, a temporary imbalance of alkalinity and $Ca^{2+}$ supplies during a transient warming event is likely when alkalinity and $Ca^{2+}$ enter the ocean via different pathways. Our set-up thus produces an upper limit for such a transient imbalance between alkalinity and $Ca^{2+}$ supply, assuming constant seafloor fluxes.

In the presented simulations, only the excursion amplitudes of radiogenic $^{87}$Sr/$^{86}$Sr, and $\delta^7 Li$ assuming climate-driven runoff modifications, are larger than measurement uncertainty in present-day seawater (see figures 3 − 6). Stronger or prolonged climate change, a larger isotopic offset between seawater and continental run-off, or a more sensitive weathering regime would be required to produce detectable weathering changes in the other isotope systems. Yet, our idealized perturbation of the modern system illustrates the differing responses of all five isotope systems to a C emission pulse. We find that the timings

of excursion peaks and minima differ strongly between the isotope systems, which might provide a 'fingerprint' of rapid C injection. For instance, the radiogenic $^{187}$Os/$^{188}$Os and negative $\delta^{88/86}$Sr and $\delta^{44}$Ca excursions peak a few tens of thousands of years after the C injection, while the radiogenic $^{87}$Sr/$^{86}$Sr and negative $\delta^7$Li excursions take hundreds of thousands of years to reach their full amplitude. These time lags result from different residence times and fractionation processes:

With the shortest residence time, estimated both from present-day inventories and in our simulation (see table 7), Os is

35 the first metal to re-equilibrate after the perturbation. The excursions in Os concentration and isotopes are only driven by

enhanced weathering rates, and thus both recover within 300 kyrs once atmospheric temperature, and with it continental weathering rates, decrease sufficiently. The radiogenic $^{87}Sr/^{86}Sr$ excursion, however, continues to grow until weathering rates have fully returned to pre-event levels because concentration-dependent Sr sinks adapt more slowly to enhanced continental Sr input as a result of the longer residence time. The re-equilibration of the marine Sr reservoir is substantially slower than

that of the Os reservoir for the same reason. However, the removal of marine Sr is also dependent on carbonate chemistry, since marine carbonate preservation constitutes the largest marine Sr sink. Biogenic carbonate preservation initially decreases because of the pH and carbonate saturation drop following the C injection, but it increases during the C cycle recovery as a result of enhanced weathering-related alkalinity input. Increased marine carbonate burial causes the marine Ca reservoir to re-equilibrate on a similar timescale to Os although Ca has a substantially longer residence time in our spin-up. The effect

is not strong enough to stop the growth of the marine Sr reservoir but it reduces its growth rate. Isotopic fractionation of Ca and Sr during biogenic carbonate formation enhances the speed of the $\delta^{44}Ca$ recovery and stops the $\delta^{88/86}Sr$ excursion from growing, despite continued excess input of isotopically light continental Sr. Li has the longest residence time of the four metals in our simulations and Li burial is only dependent on the marine Li concentration, not carbonate chemistry. Hence, the marine Li reservoir continues to grow until all excess weathering stops. Once more dissolved Li is buried than added to the ocean, i.e.

the marine Li reservoir shrinks, the $\delta^7Li$ trend reverses because excess sources of isotopically light Li ceased and isotopically light Li is preferentially buried during clay formation, enriching seawater in the heavier $^7Li$. In the longer term, stable isotope fractionation during the removal of excess Sr, Li and Ca from the ocean results in transient positive $\delta^{88/86}Sr$, $\delta^7Li$ and $\delta^{44}Ca$ excursions which, in the case of Sr and Li, last for several million years. In our simulations, the staggered timings of the $^{187}Os/^{188}Os$ and $^{87}Sr/^{86}Sr$ excursion peaks, as well as the $\delta^{44}Ca$ and $\delta^7Li$ excursion minima, are thus predominantly the result

of different elemental residence times while enhanced carbonate burial is the main reason for the time lag between $\delta^7Li$ and $\delta^{88/86}Sr$ and the coincidence of the excursion minima of $\delta^{88/86}Sr$ and $\delta^{44}Ca$.

## 5   Potential for further model development

Our Sr, Os, Li and Ca cycle representations in cGENIE constitute a useful tool to study perturbations of marine metal reservoirs during environmental changes and importantly, in concert with other biogeochemical tracers. Several extensions of this

model can be envisaged to include the represention of metal-specific processes that we did not address in this initial cGENIE implementation. Firstly, the SEDGEM module could be expanded by adding temperature dependency to the fluxes of Sr, Os, Li, Ca, C and Mg related to seafloor alteration. Although secular variation in Cenozoic climate altered these fluxes by an order of magnitude less than those connected to terrestrial weathering (Krissansen-Totton and Catling, 2017), this extra model feature could still improve the simulated metal isotope response to long-term environmental change and constrain metal fluxes

in warmer Earth system states. Secondly, the weathering module ROKGEM could be extended to simulate C and S release from organic shale weathering alongside Os fluxes to better capture the climatic and biogeochemical effects of changing shale weathering. Thirdly, celestite-bound Sr could be added as an additional particulate Sr tracer to improve the representation of Sr cycling in oceans with higher celestite stability than at present. Similarly, growth rate-dependent isotope fractionation factors

could be implemented in the ecosystem module ECOGEM (Ward et al., 2018). Yet, the benefit of more complex biological metal cycles will have to be weighed against increased computational costs.

## 6 Conclusions

Sr, Os, Li and Ca isotope records can help identify mantle activity and lithological responses to climate change in the geological record if processes governing their distribution in the ocean and their response to geological perturbations were well understood. Our implementation of the marine cycling of Sr, Os, Li and Ca in the Earth system model cGENIE allows us to investigate these processes and their consistency with observations. Simulating pre-industrial Sr, Os, Li and Ca distributions in the ocean, the model achieves slightly more homogeneous fields than observed, but consistent with the widely used assumption of homogeneous metal isotope distributions in a fully equilibrated state (e.g. Elderfield, 1986; Chan and Edmond, 1988). It reproduces the mostly homogeneous mean observed depth profiles of concentrations and isotopic compositions well, but partially deviates from local observations. This could be an artefact of measurement uncertainties, non-equilibrated cycles in today's ocean or local processes not resolved or parameterised in cGENIE. We showed how this new cGENIE tracers development provides an opportunity to investigate the consistency between reported metal compositions in the ocean and marine sources and sinks. Further investigations of the differences between cGENIE simulations and observations thus have the potential to improve our understanding of present-day marine trace metal cycling. Furthermore, the new implementation can be used to study the response of these metal cycles to transient environmental change. As an example, we showed how these systems respond to a sudden $CO_2$ release through perturbation to their input and output fluxes on different timescales. cGENIE thus becomes a valuable tool to quantitatively interrogate the environmental signals of these isotope systems as preserved in sedimentary records and their relationship to more traditional isotope systems such as carbon.

*Code availability.* The code for the version of the 'muffin' release of the cGENIE Earth system model used in this paper, is tagged as v0.9.22, and is assigned a DOI: 10.5281/zenodo.xxxxxxx.

Configuration files for the specific experiments presented in the paper can be found in https://github.com/markus-adloff/Publication$_r esources/GMD_a$

A manual detailing code installation, basic model configuration, tutorials covering various aspects of model configuration and experimental design, plus results output and processing, is assigned a DOI: 10.5281/zenodo.xxxxxxx.

## Appendix A: Sensitivity studies of the Os cycle

We tested two additional Os cycle set-ups besides the pre-industrial spin-up described in the main manuscript: one with a 2D continental weathering scheme in which Os delivery is tied to organic-rich shale and basalt weathering, and one assuming that half of marine Os deposition happens through association with POC in anoxic conditions. The following two subsections show the two set-ups and the simulation results.

## A1 Continental Os inputs calculated with a 2D weathering scheme

We added Os and stable isotope ($^{187}$Os and $^{188}$Os) tracers to the 2D weathering schemes available in ROKGEM (Colbourn et al., 2013). Derived from Suchet and Probst (1995), Bluth and Kump (1994), Gibbs and Kump (1994) and Gibbs et al. (1999), these weathering schemes calculate alkalinity and DIC fluxes entering coastal grid cells based on lithology-specific dependencies on run-off. Depending on the scheme, the model user can specify the locations of 5 or 6 pre-defined lithologies (carbonates, sandstone, shales, basalt, granites), the relative contribution of weathering fluxes from these lithologies to the overall input of carbonate- and silicate-derived weathering fluxes to the ocean and parameters that define the sensitivity of these fluxes to run-off changes. Similarly, flux and isotopic composition of Os derived from each of these lithologies can now be prescribed by changing the input files of ROKGEM (see the cGENIE.muffin manual p. 372-373 for a step-by-step guide).
To test the effect of spatially-explicit Os weathering fluxes on the isotopic composition of dissolved Os in the ocean, we tied the continental Os weathering flux to shale and basalt weathering, which are assumed to be dominant continental Os sources (Li et al., 2009). We ascribed typical isotopic compositions ($^{187}$Os/$^{188}$Os = 2.2 for shales, Dubin and Peucker-Ehrenbrink (2015), and $^{187}$Os/$^{188}$Os = 1.0 for basalt, Peucker-Ehrenbrink and Jahn (2001)) to Os derived from these lithologies and Os yields that resulted in the same global Os weathering input as in the pre-industrial spin-up presented in the main text (2605 mol Os/yr with $^{187}$Os/$^{188}$Os = 1.2).

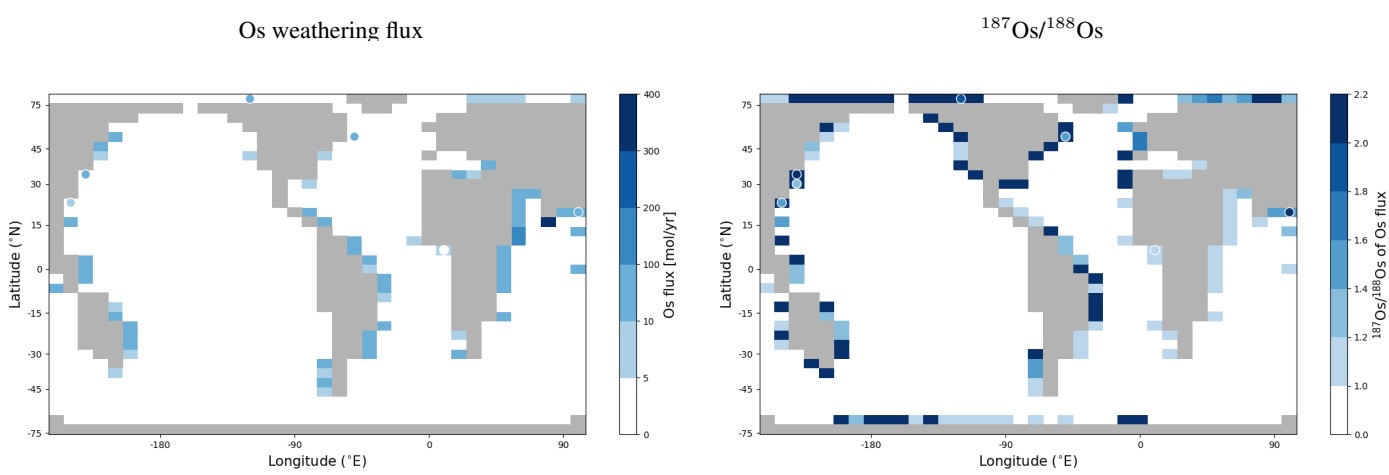

**Figure A1.** Spatially-explicit Os weathering flux using the ROKGEM 2D weathering scheme 'GEM_CO2' and observed Os river fluxes (coloured dots) based on measured Os concentrations and their isotopic composition in rivers (Levasseur et al., 1999) and annual river run-off (Milliman and Farnsworth, 2013).

Figure A1 shows the simulated spatial pattern Os weathering flux alongside measurements. The comparison between Os concentrations in rivers and in cGENIE run-off is complicated by the high reactivity of Os in river deltas (Turekian et al.,

2007), which might result in lower Os addition to the open than to the coastal ocean. Yet, there is general agreement between the measured and simulated isotopic composition of local Os weathering fluxes. This spatially heterogeneous Os weathering flux does not change the ranges of simulated marine Os concentrations and isotope ratios substantially (figure A2), suggesting that assumptions about the homogeneity of marine Os inputs are not the cause of differences between the simulated and observed marine Os reservoir.

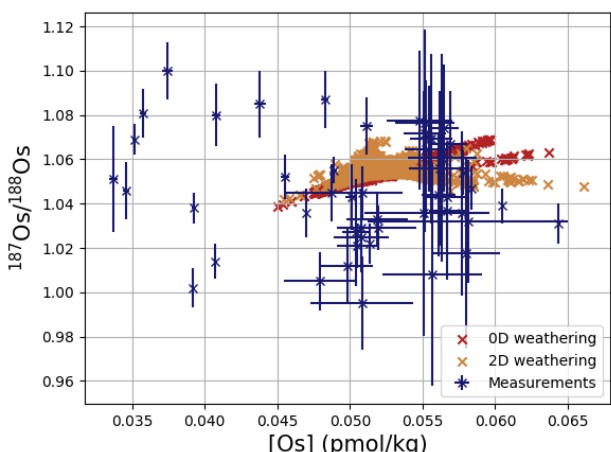

**Figure A2.** Crossplot of measured and simulated Os concentrations and isotopic compositions in seawater, comparing the modern ocean measurements against every ocean grid cell for spin ups using both the 0D and 2D weathering schemes.

## A2   Marine Os sequestration by association with POC

Another possible explanation for the wider range of observed compared with simulated Os concentrations and isotopic ratios in seawater is spatially-variable Os burial rates. The inclusion of Os cycling in cGENIE enables for the first time the simulation of the effects of heterogeneous marine biogeochemistry on Os burial. As part of the Os cycle implementation, we included options for simulating enhanced Os burial in suboxic settings and Os scavenging by POC in cGENIE, since it is assumed that low oxygen concentrations and/or burial of organic matter intensify Os sequestration (Lu et al., 2017), although rates and exact mechanisms are still uncertain. Here, we present an alteration of the previously presented spin-up of the pre-industrial Os cycle by including enhanced Os deposition via association with POC in sub-oxic conditions and scaled this sink to account for 50% of marine Os burial (see parameter choices in table A1).

**Table A1.** Model parameters setting Os scavenging under sub-oxic conditions.

| Parameter name | Model setting | Unit | Description |
|---|---|---|---|
| bg_ctrl_Os_scav_O2_dep | .true. | N.A. | switch to turn on oxygen-dependent scavenging |
| bg_par_scav_Os_O2_threshold | 5E-9 | mol kg$^{-1}$ | [O$_2$] threshold for oxygen-dependent scavenging |
| bg_par_bio_remin_kOstoPOMOS | 0 | 1 mol$^{-1}$ | scaling factor for Os scavenging |

Spatially-variable Os burial rates increase the range of simulated Os concentrations in seawater beyond that observed (figure A3). While we found that the previous range of simulated Os concentrations covered only half of the observed range, we cannot assess if the lowest simulated Os concentrations with sub-oxic Os scavenging are realistic for the pre-industrial ocean, since only a few Os measurements from oxygen-poor water masses with high POC fluxes are available. Yet, the range of measured
5   isotopic compositions of Os in seawater is also not reproduced by including spatially-variable Os burial. Further modelling and Os measurements in diverse marine settings are urgently needed to address these discrepancies.

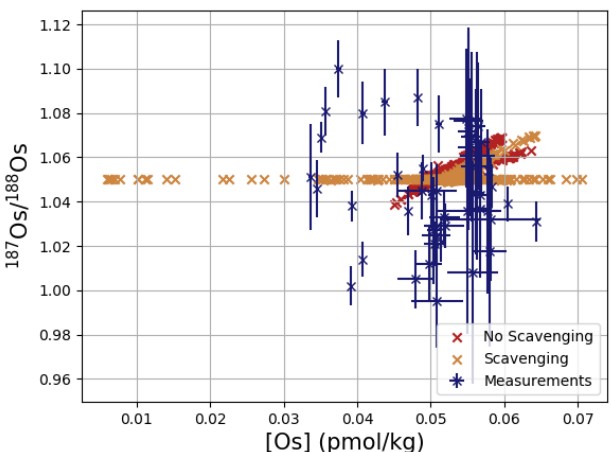

**Figure A3.** Crossplot of measured and simulated Os concentrations and isotopic compositions in seawater with and without Os scavenging under sub-oxic conditions.

## A3   Marine $^{187}$Os/$^{188}$Os signal of large C injections with O$_2$-sensitive Os scavenging

We tested the implications of dynamic Os scavenging on the isotopic response of the marine Os reservoir to climate change by repeating the previously shown C injection simulations with the Os scavenging scheme.

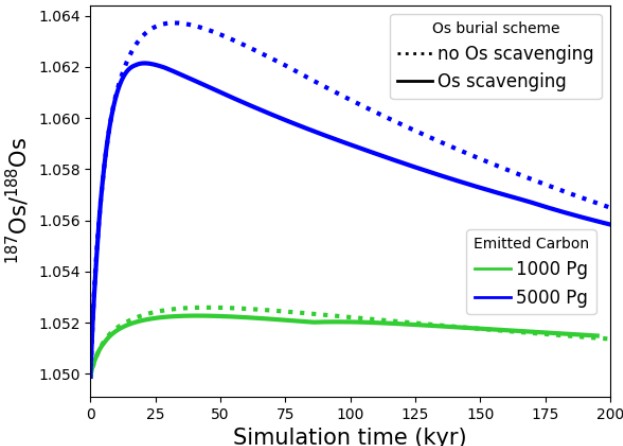

**Figure A4.** Timeseries of marine $^{187}$Os/$^{188}$Os changes in response to an immediate release of two different masses of C into the atmosphere-ocean system with and without Os scavenging from the water column

Figure A4 shows that the inclusion of Os scavenging from $O_2$-poor water masses only results in small changes to the simulated evolution of $^{187}$Os/$^{188}$Os in the C injection experiments. Enhanced remineralisation of particulate organic matter in the warmer ocean over-compensates the effects of reduced marine oxygenation and leads to less Os scavenging and hence a transient growth of the marine Os reservoir. Because the marine Os reservoir grows larger in the simulation with Os scavenging
5  than in the simulation without Os scavenging, the amplitude of the isotopic excursion is smaller.

**Appendix B: Climate-sensitive continental Li fluxes**

The abundance and isotopic composition of riverine Li depends primarily on the weathering regime and temperature during clay formation, and not the host lithology (Pogge von Strandmann et al., 2020). Simulations of continental Li fluxes based on rock weathering rates alone therefore do not capture the full variability of the Li cycle under climate change if the latter shifts the
10  weathering regime. ROKGEM does not provide a detailed land surface model, but we included the first-order approximations of the sensitivity of continental Li fluxes on the weathering regime and temperature described in section 3. Here, we show the different simulated response of the marine Li reservoir to a transient warming event with and without these approximations. For this purpose, we repeated the C injection experiments with the optional dependence of continental Li fluxes on the ratio of chemical to physical weathering rates and temperature.

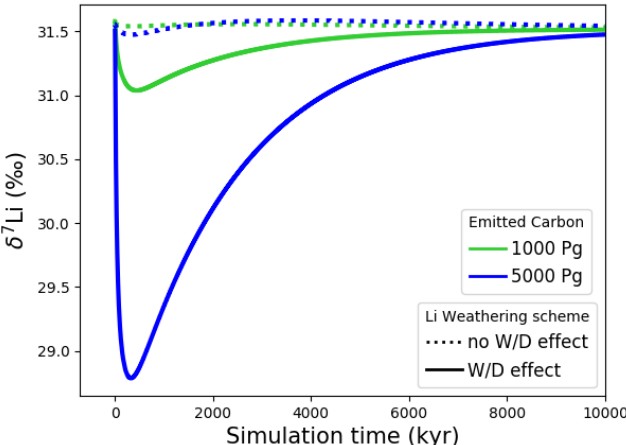

**Figure B1.** Timeseries of marine $\delta^7$Li changes in response to an immediate release of two different amounts of C into the atmosphere-ocean system with and without effects of the ratio of weathering to denudation rate (W/D) and temperature on the composition of riverine Li.

Figure B1 shows that appreciable isotopic changes in the marine Li reservoir are only simulated if the continental Li flux changes with changing $W/D$ and temperature. In the simulation with the larger C release, a sudden temperature increase, +7°C at the peak and +3°C sustained for almost 100 kyr, resulted in a marine negative $\delta^7$Li excursion of -3‰. Although our simulation is set up to represent pre-industrial metal cycles, this amplitude is at the lower end of $\delta^7$Li excursions recorded during the Paleocene-Eocene Thermal Maximum, a past C injection event with a similar global mean temperature change (Gehler et al., 2016; von Strandmann P.A.E. et al., 2019). cGENIE provides the necessary functionalities to further investigate this and other past $\delta^7$Li excursions by enabling the user to prescribe and test Early Eocene constraints on the Li cycle, including baseline weathering regimes and rates, the scale of marine clay and evaporite formation and hydrothermal inputs, and the Li concentration in seawater.

## Appendix C: Ca isotope fractionation during biogenic carbonate formation

cGENIE contains three different parameterisations for Ca isotope fractionation during carbonate formation. By default, a fixed, user-prescribed fractionation factor is applied across the whole ocean. Alternatively, the fractionation factor can be made dependent on the local concentration of carbonate ions (following Gussone et al. (2005) and Komar and Zeebe (2016)) or the local saturation state (following Tang et al. (2008). A detailed discussion of these approaches can be found in Fantle and Ridgwell (2020). Here, we show the effect of these schemes on the simulated distribution of $\delta^{44/40}$Ca in pre-industrial seawater and surface sediments.

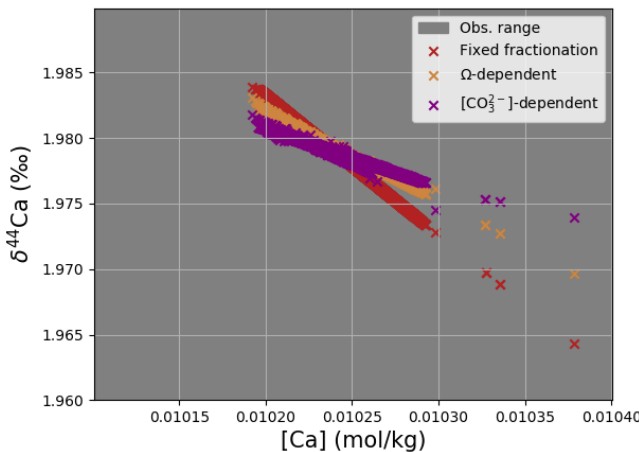

**Figure C1.** Crossplot of measured and simulated Ca concentrations and isotopic compositions in seawater using different Ca isotope fractionation schemes.

We tuned the mean seawater $\delta^{44/40}$Ca to the same value in all simulations by varying the isotopic composition of continental Ca input, hence the mean composition of seawater aligns in all three simulations but the spatial variability differs. The simulated range of $\delta^{44/40}$Ca is largest with a fixed fractionation factor because local conditions reduce the fractionation factor across most of the ocean in the simulations with dynamic fractionation factors. The smallest $\delta^{44/40}$Ca range is simulated when
5   the fractionation factor is dependent on the local concentration of carbonate ions. However, these differences are negligible compared to measurement uncertainty.

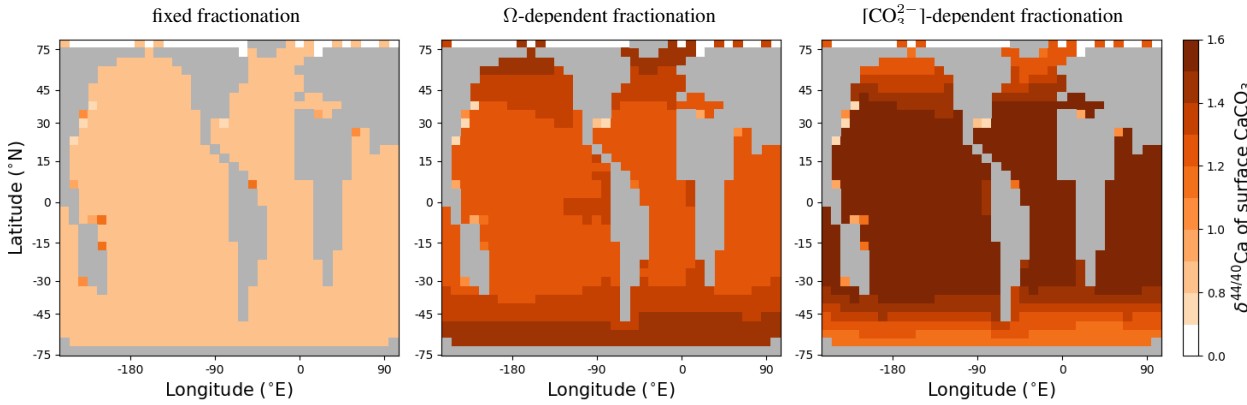

**Figure C2.** $\delta^{44/40}$Ca of CaCO$_3$ in the top layer of marine sediments in simulations with different Ca isotope fractionation schemes.

In contrast to seawater, the top layer of marine sediments shows substantial differences in $\delta^{44/40}$Ca in the three simulations (figure C2), with generally higher $\delta^{44/40}$Ca and large spatial variability when the fractionation factor is set by environmental conditions. Sedimentary carbonate $\delta^{44/40}$Ca is highest at the poles and decreases towards the equator when the fractionation factor varies with saturation state, while the opposite gradients are simulated using the carbonate-ion-dependent fractionation scheme from Gussone et al. (2005). This illustrates that the surface ocean fractionation factor is shifted in opposing directions by changing carbonate ion abundance in these fractionation schemes Fantle and Ridgwell (2020).

## Appendix D: Map of locations with prescribed reefal carbonate deposition

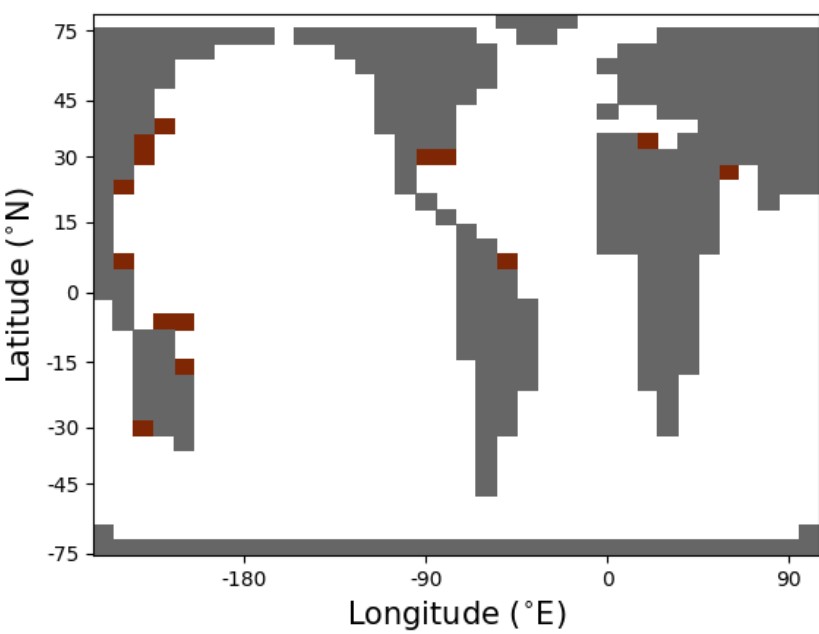

**Figure D1.** Locations with prescribed benthic carbonate deposition representing carbonate reefs. Reef grid cells are highlighted in brown.

## Appendix E: Model parameters for pre-industrial spin-up

**Table A1.** Model parameters setting continental weathering-related trace metal fluxes.

| Parameter name | Model setting | Unit | Description |
|---|---|---|---|
| rg_par_weather_CaSiO3_fracOs | 1.92E-10 | mol/mol | Os:[Ca+Mg] ratio in silicate weathering |
| rg_par_weather_CaSiO3_fracOs | 1.92E-10 | mol/mol | Os:[Ca+Mg] ratio in carbonate weathering |
| rg_par_weather_CaSiO3_fracLi | 1.33E-3 | mol/mol | Li:[Ca+Mg] ratio in silicate weathering |
| rg_par_weather_CaSiO3_fracSr | 2.7E-3 | mol/mol | Sr:[Ca+Mg] ratio in silicate weathering |
| rg_par_weather_CaSiO3b_fracSr | N.A. | mol/mol | Sr:[Ca+Mg] ratio in basalt weathering |
| rg_par_weather_CaSiO3g_fracSr | N.A. | mol/mol | Sr:[Ca+Mg] ratio in granite weathering |
| rg_par_weather_CaCO3_fracSr | 1.8E-3 | mol/mol | Sr:[Ca+Mg] ratio in carbonate weathering |

**Table A2.** Model parameters setting hydrothermal metal input.

| Parameter name | Model setting | Unit | Description |
|---|---|---|---|
| sg_par_sed_hydroip_fOs | 520 | mol yr$^{-1}$ | global hydrothermal Os flux |
| sg_par_sed_hydroip_fLi | 6E+9 | mol yr$^{-1}$ | global hydrothermal Li flux |
| sg_par_sed_hydroip_fSr | 2.86E+9 | mol yr$^{-1}$ | global hydrothermal Sr flux |
| sg_par_sed_hydroip_fCa | 0.2E+13 | mol yr$^{-1}$ | global hydrothermal Ca flux |

**Table A3.** Model parameters setting the Os, Li and Sr contents of calcite shells.

| Parameter name | Model setting | Unit | Description |
|---|---|---|---|
| bg_par_bio_red_CaCO3_OsCO3 | N.A. | mol/mol | pelagic Os:Ca ratio |
| bg_par_bio_red_CaCO3_OsCO3_alpha | 0 | unit-less | pelagic partition coefficient between Os and Ca |
| bg_par_bio_red_CaCO3_LiCO3 | N.A. | mol/mol | pelagic Li:Ca ratio |
| bg_par_bio_red_CaCO3_LiCO3_alpha | 0.02 | unit-less | pelagic partition coefficient between Li and Ca |
| sg_par_bio_red_CaCO3_LiCO3 | N.A. | mol/mol | benthic Li:Ca ratio |
| sg_par_bio_red_CaCO3_LiCO3_alpha | 0.003 | unit-less | benthic partition coefficient between Li and Ca |
| bg_par_bio_red_CaCO3_SrCO3 | N.A. | mol/mol | pelagic Sr:Ca ratio |
| bg_par_bio_red_CaCO3_SrCO3_alpha | 0.18 | unit-less | pelagic partition coefficient between Sr and Ca |
| sg_par_bio_red_CaCO3_SrCO3 | N.A. | mol/mol | benthic Sr:Ca ratio |
| sg_par_bio_red_CaCO3_SrCO3_alpha | 0.64 | unit-less | benthic partition coefficient between Sr and Ca |

**Table A4.** Model parameters setting Os scavenging.

| Parameter name | Model setting | Unit | Description |
|---|---|---|---|
| bg_ctrl_Os_scav_O2_dep | .false. | N.A. | switch to turn on oxygen-dependent scavenging |
| bg_par_scav_Os_O2_threshold | 5E-9 | mol kg$^{-1}$ | [O$_2$] threshold for oxygen-dependent scavenging |
| bg_par_bio_remin_kOstoPOMOS | 0 | 1 mol$^{-1}$ | scaling factor for Os scavenging |

**Table A5.** Model parameters for weathering-related trace metal fluxes.

| Parameter name | Model setting | Unit | Description |
|---|---|---|---|
| sg_par_sed_clay_fLi_alpha | 8.4 | kg/mol | scaling factor relating Li burial to detrital flux and [Li] |

**Table A6.** Model parameters for Sr recrystallization.

| Parameter name | Model setting | Unit | Description |
|---|---|---|---|
| sg_par_sed_SrCO3recrystTOT | 3.4E+9 | mol yr$^{-1}$ | Prescribed global SrCO3 recrystlization rate |
| sg_par_sed_SrCO3recryst | N.A. | mol cm$^{-2}$ yr$^{-1}$ | Prescribed SrCO3 recrystlization rate |

**Table A7.** Model parameters for metal deposition in sediments.

| Parameter name | Model setting | Unit | Description |
|---|---|---|---|
| sg_par_sed_lowTalt_fLi_alpha | 3.1E-8 | kg m$^{-2}$ s$^{-1}$ | Li low temperature alteration sink rate |
| sg_par_sed_lowTalt_fSr_alpha | 0 | kg m$^{-2}$ s$^{-1}$ | Sr low-T alteration sink rate |
| sg_par_sed_Os_dep | 5.9E-6 | kg m$^{-2}$ s$^{-1}$ | burial rate for Os |

**Table A8.** Model parameters for the representation of isotopes and isotopic fractionation.

| Parameter name | Model setting | Unit | Description |
|---|---|---|---|
| `rg_par_weather_CaSiO3_187Os_188Os` | 1.153 | | $^{187}$Os/$^{188}$Os of input from silicate weathering |
| `rg_par_weather_CaCO3_187Os_188Os` | 1.153 | | $^{187}$Os/$^{188}$Os of input from carbonate weathering |
| `rg_par_weather_CaSiO3_188Os_192Os` | 0.3244 | | $^{188}$Os/$^{192}$Os of input from silicate weathering |
| `rg_par_weather_CaCO3_188Os_192Os` | 0.3244 | | $^{188}$Os/$^{192}$Os of input from carbonate weathering |
| `sg_par_sed_hydroip_fOs_187Os_188Os` | 0.5625 | | $^{187}$Os/$^{188}$Os of hydrothermal input |
| `sg_par_sed_hydroip_fOs_188Os_192Os` | 0.3244 | | $^{188}$Os/$^{192}$Os of hydrothermal input |
| `bg_par_d7Li_LiCO3_epsilon` | 4.0 | ‰ | $\delta^7$Li fractionation for pelagic carbonate |
| `sg_par_d7Li_LiCO3_epsilon` | -2.0 | ‰ | $\delta^7$Li fractionation for neritic carbonate |
| `sg_par_sed_hydroip_fLi_d7Li` | 8.2 | ‰ | $\delta^7$Li of hydrothermal input |
| `sg_par_sed_lowTalt_7Li_epsilon` | -15.0 | ‰ | $\delta^7$Li of sea floor alteration sink |
| `sg_par_sed_clay_7Li_epsilon` | -15.0 | ‰ | $\delta^7$Li of clay |
| `rg_par_weather_CaSiO3_Li_d7Li` | 23.25 | ‰ | $\delta^7$Li of input from silicate weathering |
| `bg_par_d88Sr_SrCO3_epsilon` | -0.18 | ‰ | $\delta^{88/86}$Sr fractionation for pelagic carbonate |
| `sg_par_d88Sr_SrCO3_epsilon` | -0.18 | ‰ | $\delta^{88/86}$Sr fractionation for neritic carbonate |
| `sg_par_r87Sr_SrCO3recryst` | 0.7087 | | $^{87}$Sr/$^{86}$Sr of input from recrystalization |
| `sg_par_d88Sr_SrCO3recryst` | 0.256 | ‰ | $\delta^{88/86}$Sr of input from recrystalization |
| `sg_par_sed_hydroip_fSr_r87Sr` | 0.703 | | $^{87}$Sr/$^{86}$Sr of hydrothermal input |
| `sg_par_sed_hydroip_fSr_d88Sr` | 0.256 | ‰ | $\delta^{88/86}$Sr of hydrothermal input |
| `rg_par_weather_CaSiO3_r87Sr` | 0.7113 | | $^{87}$Sr/$^{86}$Sr of silicate weathering |
| `rg_par_weather_CaSiO3b_r87Sr` | N.A. | | $^{87}$Sr/$^{86}$Sr of basaltic silicate weathering |
| `rg_par_weather_CaSiO3g_r87Sr` | N.A. | | $^{87}$Sr/$^{86}$Sr of granitic silicate weathering |
| `rg_par_weather_CaCO3_r87Sr` | 0.708 | | $^{87}$Sr/$^{86}$Sr of carbonate weathering |
| `rg_par_weather_CaSiO3_d88Sr` | 0.256 | ‰ | $\delta^{88/86}$Sr of silicate weathering |
| `rg_par_weather_CaSiO3b_d88Sr` | N.A. | ‰ | $\delta^{88/86}$Sr of basaltic silicate weathering |
| `rg_par_weather_CaSiO3g_d88Sr` | N.A. | ‰ | $\delta^{88/86}$Sr of granitic silicate weathering |
| `rg_par_weather_CaCO3_d88Sr` | 0.256 | ‰ | $\delta^{88/86}$Sr of carbonate weathering |
| `rg_par_weather_CaCO3_d44Ca` | 0.80 | ‰ | $\delta^{44}$Ca of carbonate weathering |
| `rg_par_weather_CaSiO3_d44Ca` | 0.90 | ‰ | $\delta^{44}$Ca of silicate weathering |
| `sg_par_sed_hydroip_fCa_d44Ca` | 0.8 | ‰ | $\delta^{44}$Ca of hydrothermal input |
| `sg_par_d44Ca_CaCO3_epsilon` | -1.1 | ‰ | $\delta^{44}$Ca fractionation for neritic carbonates |
| `bg_par_d44Ca_CaCO3_epsilon` | -1.1 | ‰ | $\delta^{44}$Ca fractionation for pelagic carbonates |

**Appendix B: Observed and simulated metal distributions in the water column**

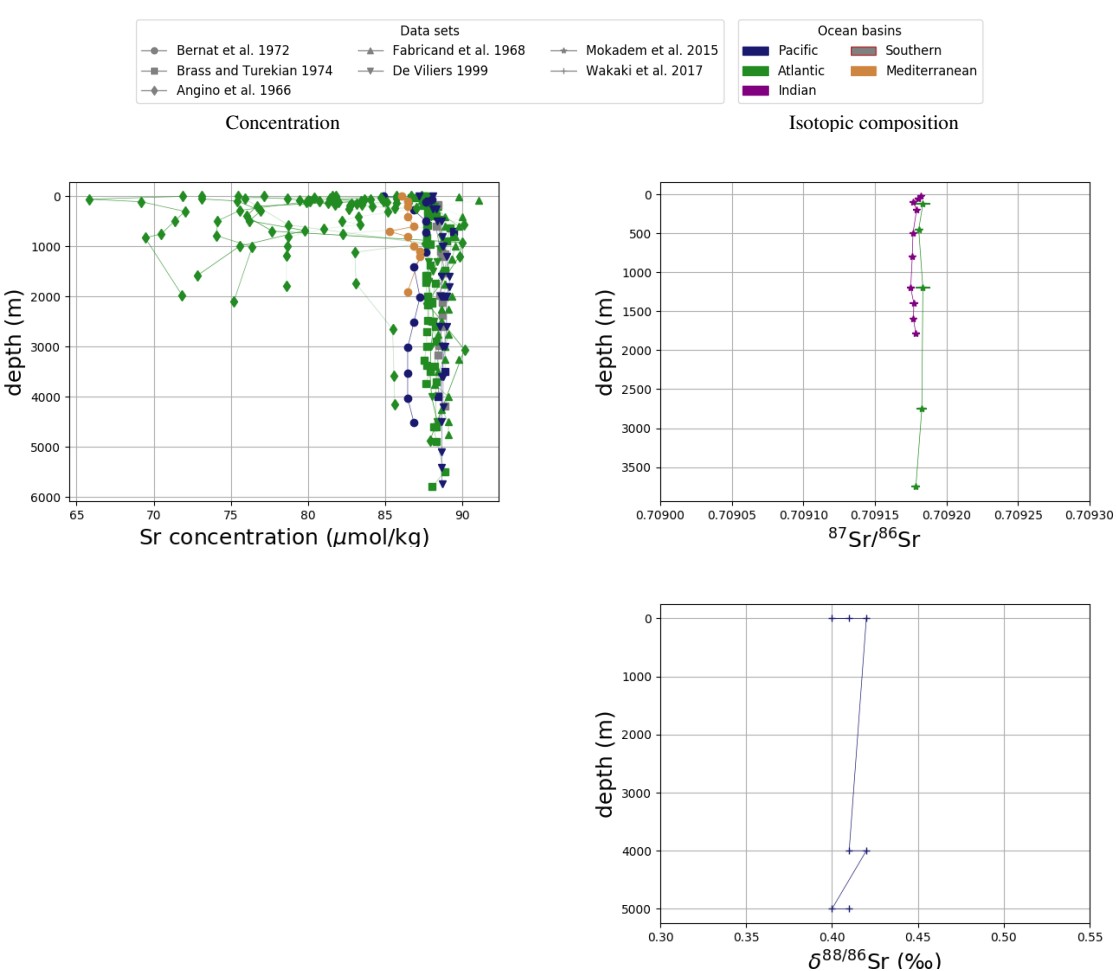

**Figure B1.** Measured Sr ocean profiles. Shown are composites of all available profiles of measured concentrations and isotopic compositions. Data are taken from Angino et al. (1966), Fabricand et al. (1967), Bernat et al. (1972), Brass and Turekian (1974), De Villiers (1999), Mokadem et al. (2015) and Wakaki et al. (2017). Sr concentrations are normalized to a salinity of 34.903495 PSU.

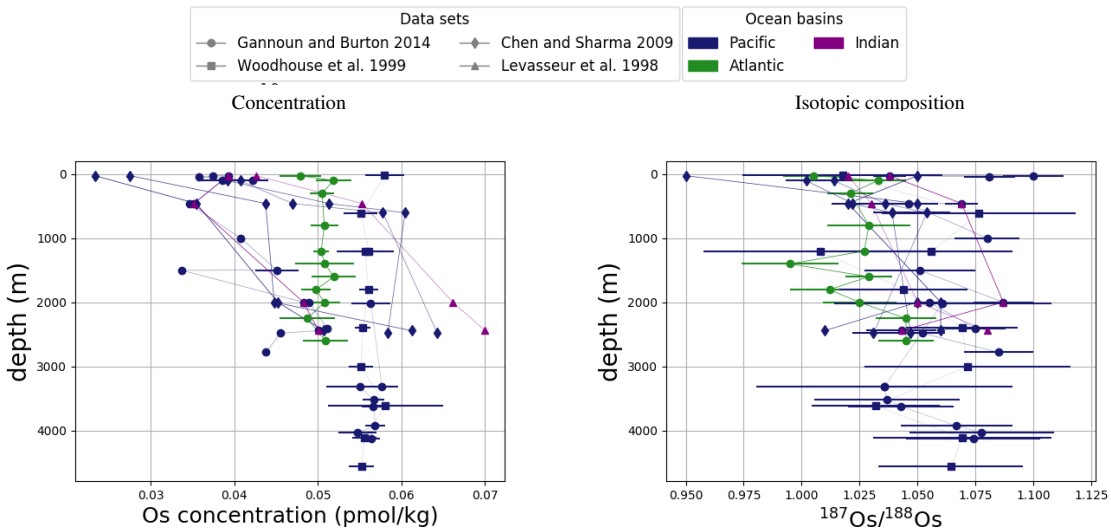

**Figure B2.** Measured Os ocean profiles. Shown are composites of all available profiles of measured concentrations and isotopic compositions. Data are taken from Levasseur et al. (1998), Woodhouse et al. (1999) and Gannoun and Burton (2014).

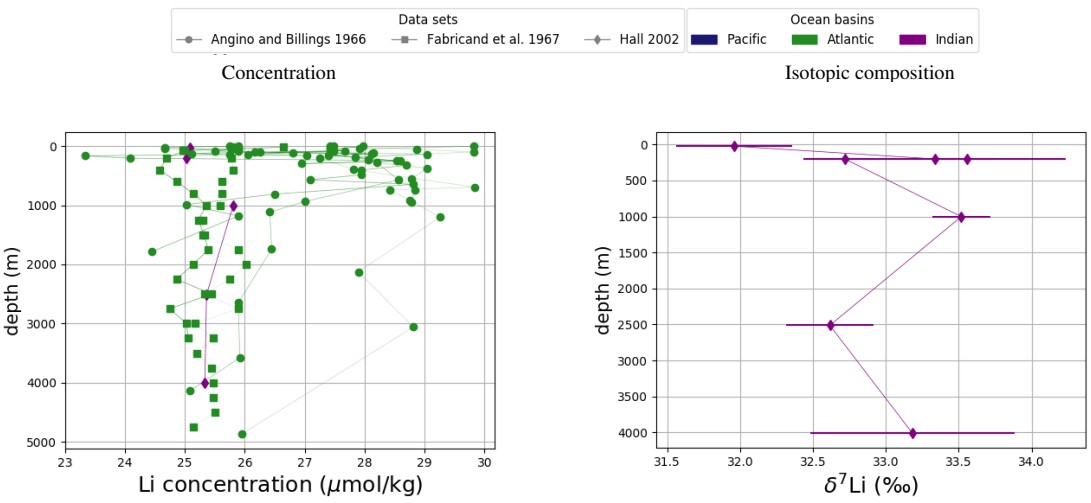

**Figure B3.** Measured Li ocean profiles. Shown are composites of all available profiles of measured concentrations and isotopic compositions. Data are taken from Angino and Billings (1966), Fabricand et al. (1967), Chan (1987) and Hall (2002). Li concentrations are normalized to a salinity of 34.903495.

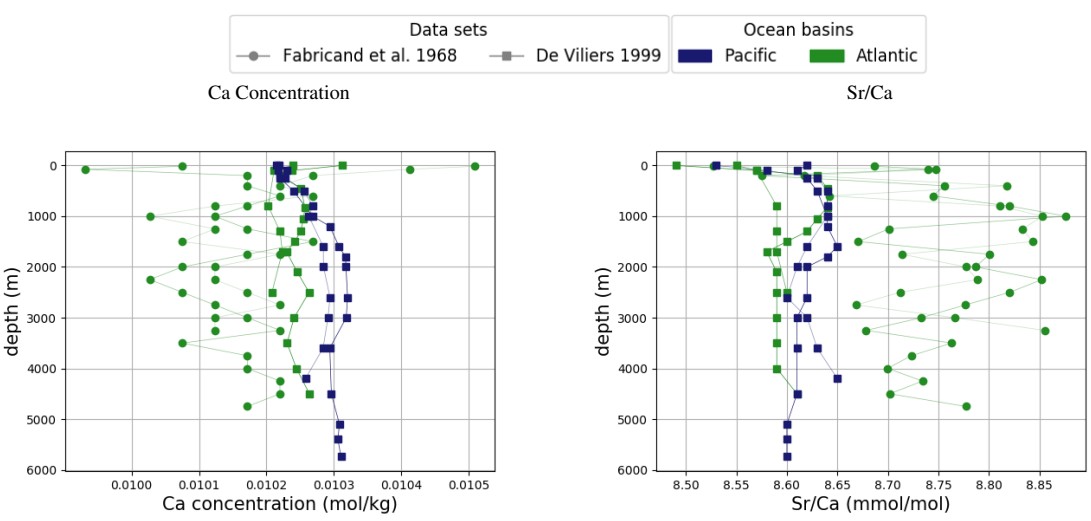

**Figure B4.** Shown are composites of all available ocean Ca concentration and Sr/Ca profiles. Data are taken from Fabricand et al. (1967) and De Villiers (1999). Ca concentrations are normalized to a salinity of 34.903495.

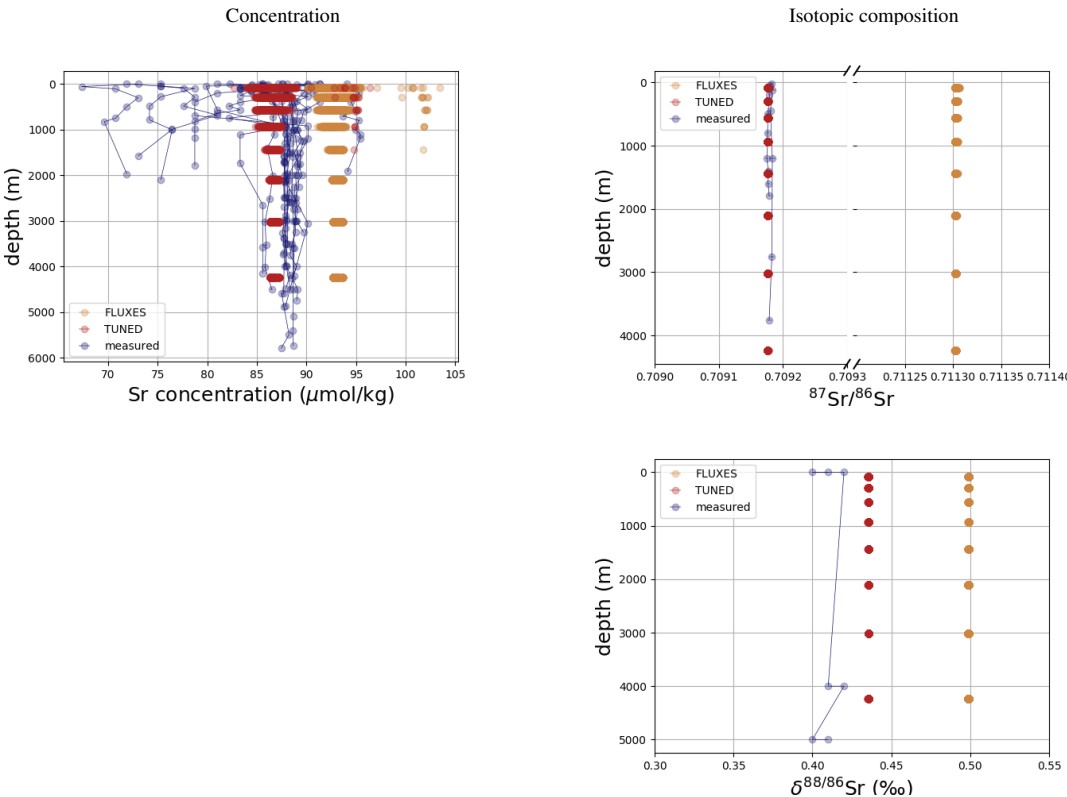

**Figure B5.** Comparison between measured and simulated Sr profiles in seawater. Shown are composites of all available measured Sr concentration and isotope ratio profiles and all profiles in the simulation. Data is taken from Angino et al. (1966), Fabricand et al. (1967), Bernat et al. (1972), Brass and Turekian (1974), De Villiers (1999) and Mokadem et al. (2015). Sr concentrations are normalized to a salinity of 34.903495.

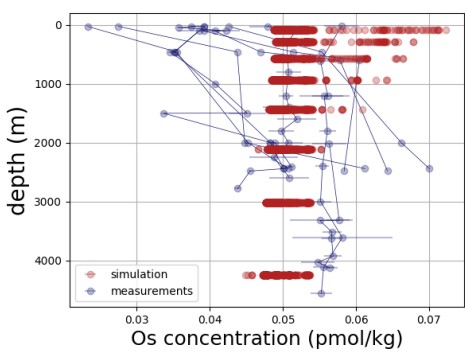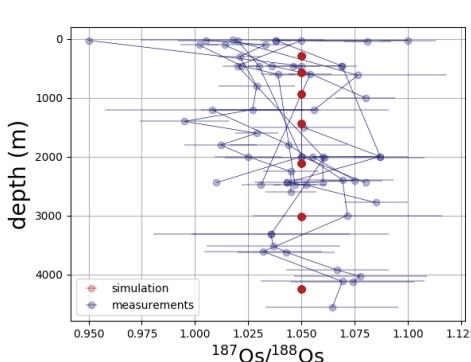

**Figure B6.** Comparison between measured and simulated Os vertical profiles in seawater. Shown are composites of all available measured Os concentration and isotope ratio profiles and all profiles in the simulation. Data is taken from Levasseur et al. (1998), Woodhouse et al. (1999) and Gannoun and Burton (2014).

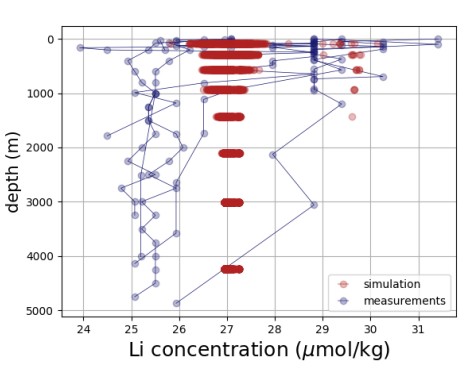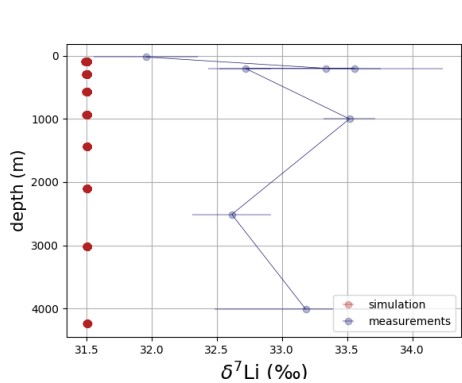

**Figure B7.** Comparison between measured and simulated Li profiles in seawater. Shown are composites of all available measured Li concentration and isotope ratio profiles and all profiles in the simulation. Data is taken from Angino and Billings (1966), Fabricand et al. (1967), Chan (1987), Chan and Edmond (1988), You and Chan (1996), Moriguti and Nakamura (1998), Tomascak et al. (1999), James and Palmer (2000), Košler et al. (2001), Nishio and Nakai (2002), Hall (2002), Bryant et al. (2003), Pistiner and Henderson (2003), Millot et al. (2004), Choi et al. (2010) and Lin et al. (2016). Li concentrations are normalized to a salinity of 34.903495.

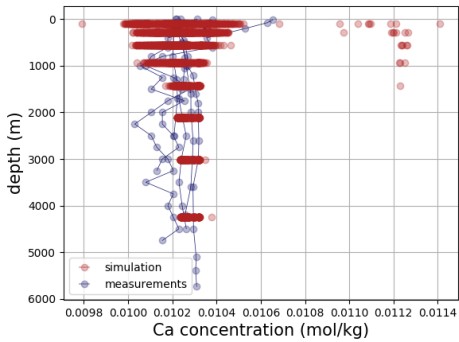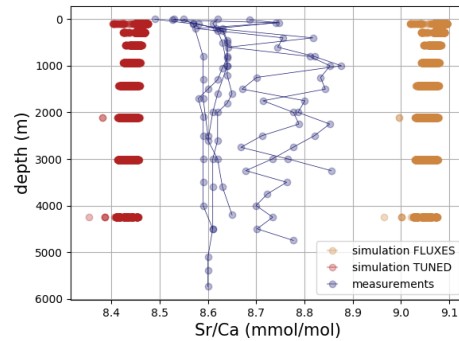

**Figure B8.** Comparison between measured and simulated Ca and Sr/Ca profiles in seawater. Shown are composites of all available measured concentration profiles and all profiles in the simulation. Data is taken from Fabricand et al. (1967), De Villiers (1999) and Fantle and Tipper (2014). Ca concentrations are normalized to a salinity of 34.903495.

## Appendix C: Site specific model-data comparison

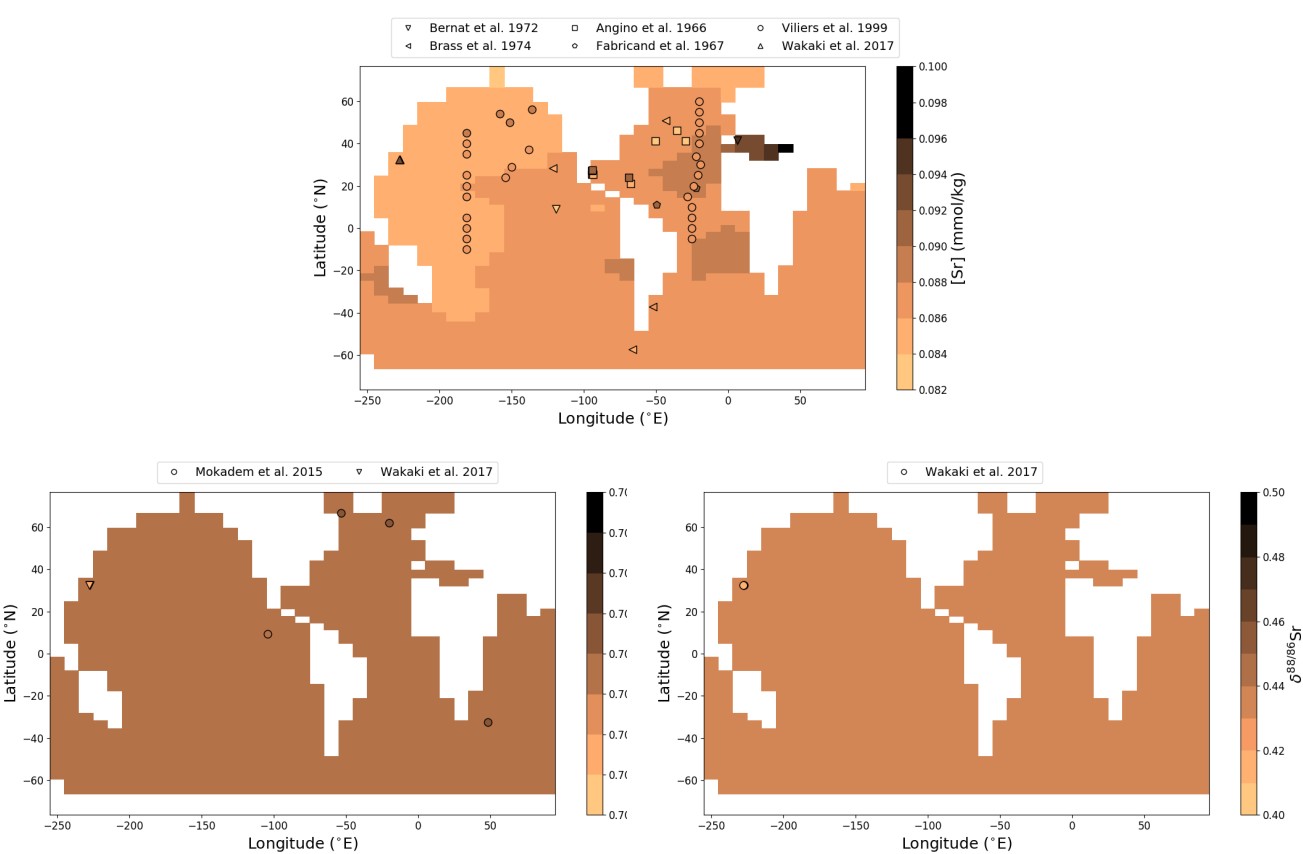

**Figure C1.** Comparison between measured and simulated Sr concentrations and isotopic composition in surface waters. Measurements are shown as symbols on the map, the colour indicating the respective value. Data is taken from Angino et al. (1966), Fabricand et al. (1967), Bernat et al. (1972), Brass and Turekian (1974), De Villiers (1999), Mokadem et al. (2015) and Wakaki et al. (2017).

## Appendix D: Seawater measurements used for the model evaluation

Summary tables of all Sr, Os, Li and Ca concentration measurements, their locations and references are provided as separate spreadsheets. Ca isotope ratios in seawater were taken from the compilation in Fantle and Tipper (2014).

5 *Author contributions.* The implementation of the Sr, Li and Ca cycles was conceived by AR, PPvS and MF and done by AR. The cGENIE Os cycle was conceived by MA, IP, AD and AR and implemented by MA and AR. MA compiled the observational data, validated the model output and produced the simulations and figures. MA prepared the manuscript with contributions from all co-authors.

*Competing interests.* No competing interests are present.

*Acknowledgements.* We would like to thank our funding bodies: M.A. was supported by the NERC GW4+ DTP and the NERC grant NE/L002434/1. S.E.G. was supported by NERC grants NE/L011050/1 and NE/P01903X/1 while working on this manuscript. FMM was supported by a NERC standard grant (NE/N011112/1). PPvS was supported by ERC Consolidator grant 682760. AR was supported by NSF grants 1658024 and 1702913 and by the Heising Simons Foundation.

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
