# Peer review of "Inclusion of a suite of weathering tracers in the cGENIE Earth System Model - muffin release v.0.9.22"

_Geoscientific Model Development, 2020_

## Referee Comment (RC1) · Anonymous Referee #1 · 19 Oct 2020

I read this paper with great interest- cGENIE is already a hugely important model that has been central to a vast number of advances in understanding our Earth's climate, and the controls that influence it. As silicate weathering is an important component of climate regulation, it makes a lot of sense to incorporate these weathering-sensitive isotope systems into the model- and in doing so it opens all sorts of possibilities: for example to reverse model past events where these isotope records are published and available. As such this model is a really important and exciting advance in the field. There is no question that a model description paper such as this is suitable for this journal, and the material is clearly significant enough and important enough to warrant publication. I have a number of suggestions, however, that I think should be addressed

in order to improve the model and its presentation in this manuscript.

There are a couple of issues with the model that I am not sure about- in particular with how Os is being handled. For instance, there is no Os input from Corg weathering, but output into POC and COrg burial. This to me seems a bit of a shame- partly the system seems underparameterised, but more importantly also, it really limits the model's potential in evaluating some important hypotheses in Earth history regarding weathering changes and its effect on the carbon cycle. For instance, one cannot test to what extent Os isotopes were reflecting just the exhumation of new lithologies high in Os (see Myrow et al. 2015, EPSL), vs. weathering changes, because one cannot change the lithological map. See also things worth testing with such a spatially detailed and complex model in papers by Zhang & Planavsky (Am. J. Sci., 2019) and Jagoutz et al. (2016, PNAS). It seems to me reading section 3.1 that although there is no explicit representation of organic matter weathering in ROKGEM, and sure carbonate-vs-silicate rock would not be a very good bounding line for modelling Os, there is a representation of shale lithologies in ROKGEM. Given the importance of shale in Os weathering fluxes, couldn't the model at least try to represent lithology in a more mechanistic way that would allow better flexibility in terms of what could be modelled? It seems a bit of a cop-out/missed opportunity to just scale to continental runoff, when the system is so sensitive to the lithology of the Earth surface..

Also with Os, I was disappointed by the decision taken in Section 4.2.2 to ignore data-model mismatch on the basis that it 'should be the basis of a separate study'.. isn't the whole point of this paper to be the paper that presents a working model? If there is a fundamental process affecting Os distribution that is missing in the model such that it can't replicate the modern, shouldn't that tell us that it isn't ready to apply to the past? It isn't really an issue of only being interested in basin-scale patterns if there is a missing, possibly unknown, process somewhere that could have been far more prominent in the past and thus render the whole Os representation inaccurate. There is some mention on Page 24, Line 17 of it being due to Os binding to organic matter in the water column

in low O2. This is confusing, because in Section 3.5 there is a rough parameterization of this process – can't the model just include an [O2]-dependent sink here? Given the propensity of low O2 regions during OAEs, mass extinctions, etc. where this model might be used, isn't getting this process right of utmost importance?

Secondly, the presentation of the data/model fits could be improved in places:

- In Fig. 1, what are the grey shaded regions, and how are they derived? This is not explained. And do the means and shaded regions take into account the uncertainty on each data point, in some sort of Monte Carlo fashion? It might be advisable- for example with Li where some of the extreme values have reasonably high uncertainty, and so should not be weighted in the same way when calculating the mean.

- I appreciate that the authors should not be expected to critically examine the methodologies and structural sources of uncertainty in all these data publications. However, there might be some simple ways to help the reader ascertain which of these data should be considered more reliable as estimates of the seawater isotope composition or element concentration.. For instance, in recognition of the advances in mass spectrometry in the past decades that result in superior analytical accuracy and precision relative to some of these pioneering but now perhaps less trusted estimates, might it be reasonable to colour-code the measured data by year published? Or if that is too busy, one could decide on an arbitrary cut-off date (e.g. 2000) and draw the data points before that date as slightly lighter than the more recent, and likely more reliable, estimates? Of course, chronology is an imperfect metric of reliability, but it is likely to be at least indicative (I noticed that the datasets that are most inclined to diverge from the model are often the older studies; e.g. Angino 1966, measured via atomic absorption spectroscopy). It's easier to evaluate the model's performance if the reader has a sense which datapoints might be most likely to be reliable.

- In Fig. 2, the yellow Calcium arrows are very difficult to see at times.. could use a different colour- sky blue?

- I can see how these plots ranking measurements in terms of values is sensible and meaningful for isotope systems where the residence time in seawater is vastly more than the timescales of circulation.. But in the case of Os, where there is a chance of some regional differences, it doesn't make so much sense. Because if the spread in the data is as shown but actually most of the data is all from one region, say, like the North Atlantic, but the model has every grid point in the ocean, you're comparing apples and oranges. The model extremes might reflect spatial differences, but the data extremes might reflect unreliable measurements but from a limited geographic range. But the plot makes it seem like the comparison should be meaningful. Something more like Figs. A3 or C1 would likely be more useful, but then at the same time these figures are not as helpful as they could be. In Fig. A3 it looks like Os concentrations and isotopes are totally homogenous everywhere in the model. But in Fig. 5 there are clearly divergent values that suggests Os in seawater is not homogenous. So which way is it? In Fig. A3 also there are few enough measured profiles that you could give them different symbols for different ocean basins, and a colour that scales with latitude?

- A lot of the figures are very very small and awkward to read. For instance Fig. A1.. this is a lot of panels of different sizes and shapes, about different things, and thrown together in one figure- some with superimposed labels in a jarring serif font positioned in an odd way, some without labels, etc.. Why not just make them separate supplementary figures for each isotope system - there is no limit on numbers I would guess, and then you could have plots that people can read properly.

- Shouldn't one of the plots in Figure 5 be concentration?

- In Fig. 7 the light and dark lines are hard to tell apart- particularly the orange, purple and green. Can't one be dotted and one be dashed, as well?

I find the examination of the weathering response to CO2 is very interesting, but also very surprising. If the model suggests that an instant release of as much as 5,000 Pg of C instantaneous results in barely any change in d7Li, but the PETM, where a similar

order of pCO2 was released saw a 3-4 ‰ change in d7Li (Pogge von Strandmann P, Jones M, Schmidt D & Murphy M (2019) Goldschmidt Abstracts, 2019 2682), then what gives? Is the model undersensitive, or missing a flux? Or is the Li isotope excursion at the PETM caused by something else? It would be good to discuss this, because otherwise it is impossible to tell how much of it is due to missing fluxes/reactions, or to know what they might be.

In Tables 1-4, values for various parameters and estimates are given, with references for the source in each case. However, from Table 5 on, when talking about model default parameters, there is no such detail. It can be hard then for the reader to know which values are chosen for a good scientific reason, and which are chosen just based on not wanting the model to crash. It would be good to say where these numbers came from. Also it seems like a disproportionately high number of parameters are set to zero- is this because these are the choices made in the chosen scenario only, or are these a lot of parameters that are theoretically included but that people should not use because they may crash the model?

Page 28, Line 26: Would it be hard or computationally-expensive to just scale the Ca2+ source at the seafloor to the bottom water temperature that is simulated?

Section 3.1: I know there are descriptions in the ROKGEM paper, but a cursory explanation of how source rock lithologies are programmed into the model would be helpful in this paper, so the reader doesn't have to read a whole other paper to understand this one.

Section 3.7: Should strontium sulphate delivery by Acantharians matter and be considered?

Page 2, Line 5: mantle spelt wrong.

Page 6, Line 13: Perhaps Dellinger et al. (2015, GCA) might be a good paper to include as a citation here?

[Figure]

Page 9, Line 9: I would say it's due to analytical inaccuracy, technically, rather than the uncertainty bounds on the values.

Page 12, line 21: Would it be possible to give the user a knob to turn to partition more or less hydrothermal flux in one ocean basin vs. another at least? Thinking of the Cretaceous with restricted seaways- Os may be heterogenous, and centres of hydrothermal activity also.

Page 13, Line 19: Why only Li in authigenic carbonates, but not others like Sr?

Page 14, Line 12: buried spelt wrong. Also a reference for the statement would be good.

Page 18, Table 13: I get that there is no fractionation of Os isotopes parameterized, but this description in this table suggests that the 187/188Os signature of inputs is set to 0, which seems to be wrong..?

Page 19, Lines 21-22: Why is this needed to balance the Ca cycle? And what is the natural process this is supposed to mimic? Can you give a reference?

Page 24, Line 15: not sure 'against' is the right word to use here.

Page 26, Line 1: no date for the Hall reference

Page 26, Line 11-12: Say 'another' indication that this paper is wrong.. but wasn't the first point of discussion earlier in the paper that these values could be wrong talking about Angino and Billings (1996), not Angino et al.?

Fig. A1: Never heard of the Indic Ocean! Also see my main gripe. Aesthetics in these plots could also be a lot nicer- it's very default python. For instance the colour palette for the data series or the data point symbols could be used to convey information about ocean basin, or study, or something.. The Os isotope panel for example is really hard to read and not nice on the eyes.

Fig. A2-A5: Many of these figures really don't print well. The various greys and blacks

are often very hard to tell apart when printed- the FLUXES vs TUNED for example. I would using some colours to help with this. Fig A2 has a mixture of [] and () in axis titles. Numbers on the axes are unnecessarily small.

Fig C1: These datapoints are far too small! Impossible to judge the colour of the points. Ditto the axis labels and legend entries.. There is a missing colour scale label for the bottom left panel, and the X longitude axis is completely cut off for the bottom two panels.

Acknowledgments: For some reason starts with 'Furthermore'.. Are there some sentences missing/lost before?

―――――――――――――――

---

## Referee Comment (RC2) · Anonymous Referee #2 · 5 Nov 2020

The contribution by Adloff et al. describes a new module of the cGENIE model, devoted to the calculation of the elemental and isotopic cycling of several trace metals (namely Sr, Ca, Li and Os).

The paper is globally organized in three sections: (1) a brief but very well written review of the trace metal behavior in the surficial Earth System, (2) a description of the implementation of those processes, and (3) results for the present day configuration and a test exploring the response of the trace metals (including their isotopic signature) to a transient perturbation.

Overall, the paper is very well organized. From my reading, I have three comments.

[Figure]

As clearly mentioned and explained in the paper, the cycling of the trace metals (TM) and their isotopic signatures is heavily dependent on the continental weathering processes, in a way making them difficult to simulate. Here the authors choose to link the trace metal fluxes to the weathering flux as calculated by another cGENIE module (ROKGEM). The authors assume that TM fluxes are proportional to the silicate and/or carbonate weathering fluxes. This is probably true for Sr and Ca, but not for Li and Os. Os is heavily dependent on the presence of particulate organic matter. Continental Li fluxes and their isotopic signature are controlled by the interactions between secondary phases and the continental waters. Those processes are not included in the model. Typically, periods of intense weathering will be characterized by a large retention of Li inside secondary minerals, strongly reducing the Li fluxes during the phase of regolith growth, while major cation fluxes released by the weathering of fresh rocks will be at a high level (Vigier and Goddéris, 2014). Li fluxes and isotopic signature are further impacted by the presence of large flooded areas on the continents, where continuous exchanges with clay minerals can occur, directly impacting the isotopic signature of the continental Li discharge (Dellinger et al., 2015; Maffre et al., 2020). I'm not saying that it is mandatory to include all this in a global scale model, but I fear a bit that the proportionality hypothesis will generate wrong interpretations of the seawater isotopic signal. This should be discussed in the paper, for example by stating that this is a first step towards something closer to the physics of the weathering system.

My second point is related to the residence times of the TM. A clearly stated by the authors, the residence times of the fourth considered TM in the ocean is much longer than the mixing time of the ocean. This implies that the seawater signal will be uniform all around the world. This is shown on figure 1. So why using a complex 3D oceanic model ? The mixing will generate a uniform distribution of the TM and their isotopic signature. The interest of including those TM in the model is more to constrain the contribution of the continental and oceanic crust weathering to the flux of elements. If the objective is to constrain the oceanic mixing, elements displaying shorter residence time are best fitted, such as the Nd. This said, the long residence time does not pre-

clude spatial variations, especially on continental margins as it is the case for strontium (El Meknassi et al., Geology, 2020). But those margins are generally not represented in the models. So, I'm just wandering whether a better description of the objectives should appeared or not ? (I would say yes).

My last point is related to the implementation of the runoff. I checked in the Colbourn contribution, but I was not able to understand precisely how it works. A brief description should appear in the text. In summary, this contribution is valuable and should be published, with the above points clarified.

---

## Author Comment (AC1) · 12 Jan 2021

Inclusion of a suite of weathering tracers in the cGENIE Earth System Model – muffin release v.0.9.10

Responses to reviewer comments

On behalf of the author team, I thank anonymous reviewers #1 and #2 for their consideration of our manuscript, their comments and ideas for improvements. We appreciate the in-depth discussion of our model and manuscript by both reviewers and propose a wide range of model and manuscript changes based on their suggestions. Below we

[Figure]

respond to each comment and describe how we will address the raised concerns in a revised version of the manuscript. I also attached a colour-coded version of our replies (blue - review comment, black - our reply) as a pdf to this comment, in case that this is easier to read.

Replies to Reviewer #1

Comment 1: there are a couple of issues with the model that I am not sure about- in particular with how Os is being handled. For instance, there is no Os input from Corg weathering, but output into POC and COrg burial. This to me seems a bit of a shame- partly the system seems underparameterised, but more importantly also, it really limits the model's potential in evaluating some important hypotheses in Earth history regarding weathering changes and its effect on the carbon cycle. For instance, one cannot test to what extent Os isotopes were reflecting just the exhumation of new lithologies high in Os (see Myrow et al. 2015, EPSL), vs. weathering changes, because one cannot change the lithological map. See also things worth testing with such a spatially detailed and complex model in papers by Zhang & Planavsky (Am. J. Sci., 2019) and Jagoutz et al.(2016, PNAS). It seems to me reading section 3.1 that although there is no explicit representation of organic matter weathering in ROKGEM, and sure carbonate-vs-silicate rock would not be a very good bounding line for modelling Os, there is a representation of shale lithologies in ROKGEM. Given the importance of shale in Os weathering fluxes, couldn't the model at least try to represent lithology in a more mechanistic way that would allow better flexibility in terms of what could be modelled? It seems a bit of a cop-out/missed opportunity to just scale to continental runoff, when the system is so sensitive to the lithology of the Earth surface..

Our Reply: We agree with the reviewer that tying Os concentration and isotopic composition in run-off to total weathering rates is a simplification that should be tested. While ROKGEM can be used to calculate globally-averaged (0D scheme) and spatially-explicit (2D scheme) weathering fluxes, the former – which only distinguishes between carbonate and silicate rock weathering – has been extensively published with in simulating weathering feedback on carbon release and for this reason and traceability, we focused on that scheme in this paper. However, and particularly in light of the observed variance in model ocean Os concentrations that is not simulated under our current (mean global weathering) assumptions, we will test and discuss the consequences of more resolved representations of Os weathering. Specifically:

Planned model improvements: Firstly, we will provide separate parameters for the concentration and isotopic composition of Os derived from carbonates (i.e. linear temperature dependence) and from two adjustable fractions of silicates (i.e. exponential temperature dependence), one of which will be used to represent the higher concentration and radiogenic nature of Os in shales. Secondly, we will configure ROKGEM in a fully 2D mode including an explicit representation of the modern distribution of shale rock types. We will use this to test whether accounting for geographical differences in shale occurrence leads to any improvement in the model simulated oceanic distribution of Os concentrations (and isotopic values).

Planned manuscript improvements: We will show marine Os distributions with the 0D and 2D weathering schemes in the supplementary material, and describe the two weathering schemes more thoroughly in the main manuscript.

Comment 2: Also with Os, I was disappointed by the decision taken in Section 4.2.2 to ignore data-model mismatch on the basis that it 'should be the basis of a separate study'.. isn't the whole point of this paper to be the paper that presents a working model? If there is a fundamental process affecting Os distribution that is missing in the model such that it can't replicate the modern, shouldn't that tell us that it isn't ready to apply to the past? It isn't really an issue of only being interested in basin-scale patterns if there is a missing, possibly unknown, process somewhere that could have been far more prominent in the past and thus render the whole Os representation inaccurate. There is some mention on Page 24, Line 17 of it being due to Os binding to organic matter in the water column in low O2. This is confusing, because in Section 3.5 there is a rough parameterization of this process – can't the model just include

an [O2]-dependent sink here? Given the propensity of low O2 regions during OAEs, mass extinctions, etc. where this model might be used, isn't getting this process right of utmost importance?

Our Reply: We agree with the reviewer that it is important to understand the sink mechanisms of Os before interpreting Os records across past intervals. At present, the exact pathway of Os from solution to sedimentary sinks is not known. Different theories exist, including but not limited to association with organic matter and correlation with dissolved O2 concentrations in the sediments or the water column (Woodhouse et al. [1999], Gannoun et al. [2014]). For this reason, our initial Os modeling did not include a mechanistic sink for dissolved Os but instead removed a constant fraction of the dissolved Os inventory, comparable to previous Os cycle models (e.g. Tejada et al. [2009]). (A full exploration of the alternative mechanisms and sinks for Os was in fact intended for a follow-up paper.) However, we will now include an explicit model test and discussion of the effect of including an anoxic scavenging sink for Os (in addition to the current generic (oxic) lifetime-based removal term). This will be done in conjunction with the test of an explicit 2D weathering field to help elucidate the reasons for the 'data-model mismatch'.

Planned manuscript improvements: We will add a comparison and discussion of Os concentrations simulated with a diffusive sink and with anoxic Os scavenging by organic particles to the supplementary material of the present manuscript. We will also provide an example of if and how this modulates the dynamical response of Os in the ocean to massive carbon release.

Comment 3: In Fig. 1, what are the grey shaded regions, and how are they derived? This is not explained. And do the means and shaded regions take into account the uncertainty on each data point, in some sort of Monte Carlo fashion? It might be advisable- for example with Li where some of the extreme values have reasonably high uncertainty, and so should not be weighted in the same way when calculating the mean.

Our Reply: The grey shaded regions currently show +/- one standard deviation around the arithmetic mean of all measurements.

Planned manuscript improvements: We will follow the reviewer's suggestion and weigh measurements by their reported uncertainty in the revised manuscript. We will also add an explanation of the grey shading to the legend.

Comment 4: I appreciate that the authors should not be expected to critically examine the method-ologies and structural sources of uncertainty in all these data publications. However, there might be some simple ways to help the reader ascertain which of these data should be considered more reliable as estimates of the seawater isotope composition or element concentration.. For instance, in recognition of the advances in mass spectrometry in the past decades that result in superior analytical accuracy and precision relative to some of these pioneering but now perhaps less trusted estimates, might it be reasonable to colour-code the measured data by year published? Or if that is too busy, one could decide on an arbitrary cut-off date (e.g. 2000) and draw the datapoints before that date as slightly lighter than the more recent, and likely more reliable, estimates? Of course, chronology is an imperfect metric of reliability, but it is likely to be at least indicative (I noticed that the datasets that are most inclined to diverge from the model are often the older studies; e.g. Angino 1966, measured via atomic absorption spectroscopy). It's easier to evaluate the model's performance if the reader has a sense which datapoints might be most likely to be reliable

Our Reply: We share the reviewer's concerns about the possibility of under-estimated analytical uncertainties for some of the published measurements, although it is beyond the scope of a model-development paper to quality-control available observed data for 4 different metals species and 5 isotopes systems. However, we can improve the data provision and include information such as publication year that can be entrained in the manuscript discussion.

Planned manuscript improvements: In supplementary material – we will colour-code

the decade in which measurements were published and provide a full list of the plotted data points with references.

Comment 5: In Fig. 2, the yellow Calcium arrows are very difficult to see at times.. could use adifferent colour- sky blue?

We will change the colour decoding Calcium pathways to blue as suggested.

Comment 6: I can see how these plots ranking measurements in terms of values is sensible and meaningful for isotope systems where the residence time in seawater is vastly more than the timescales of circulation.. But in the case of Os, where there is a chance of some regional differences, it doesn't make so much sense. Because if the spread in the data is as shown but actually most of the data is all from one region, say, like the North Atlantic, but the model has every grid point in the ocean, you're comparing apples and oranges. The model extremes might reflect spatial differences, but the data extremes might reflect unreliable measurements but from a limited geographic range. But the plot makes it seem like the comparison should be meaningful. Something more like Figs. A3 or C1 would likely be more useful, but then at the same time these figures are not as helpful as they could be. In Fig. A3 it looks like Os concentrations and isotopes are totally homogenous everywhere in the model. But in Fig. 5 there are clearly divergent values that suggests Os in seawater is not homogenous. So which way is it? In Fig. A3 also there are few enough measured profiles that you could give them different symbols for different ocean basins, and a colour that scales with latitude?

Our reply: The figures were intended to enable an easy assessment of the homogeneity of metal concentrations and isotopes in observations and simulations but we recognize that they can be misleading in case of sampling bias and do not enable a detailed discussion of differences between observations and simulations. Fig. A3 erroneously showed average concentrations instead of all grid cell values.

Planned manuscript improvements: We will provide completely-revised model-data

comparison figures utilizing very different visualization to enable a more meaningful and accessible discussion of discrepancies between model output and measurements. We will also correct Fig. A3 accordingly and combine the Os-related panels from the original Fig. A1 with Fig. A3, so that data points can more easily be traced back to the study they were reported in.

Comment 7: A lot of the figures are very very small and awkward to read. For instance Fig. A1..this is a lot of panels of different sizes and shapes, about different things, and thrown together in one figure- some with superimposed labels in a jarring serif font positioned in an odd way, some without labels, etc.. Why not just make them separate supplementary figures for each isotope system - there is no limit on numbers I would guess, and then you could have plots that people can read properly.

We will change the arrangement of figures so that there is one figure for each element with panels displaying the measured vertical profiles.

Comment 8: Shouldn't one of the plots in Figure 5 be concentration?

Our Reply: Yes, and the same issue occurred in Figure 1.

Planned manuscript improvements: We will correct both figures so that they show concentrations and isotopic composition of dissolved Li.

Comment 9: In Fig. 7 the light and dark lines are hard to tell apart- particularly the orange, purple and green. Can't one be dotted and one be dashed, as well?

One line will be dotted, as suggested by the reviewer.

Comment 10: I find the examination of the weathering response to CO2 is very interesting, but also very surprising. If the model suggests that an instant release of as much as 5,000 Pgof C instantaneous results in barely any change in d7Li, but the PETM, where a similar order of pCO2 was released saw a 3-4 ‰ change in d7Li (Pogge von Strandmann P,Jones M, Schmidt D & Murphy M (2019) Goldschmidt Abstracts, 2019 2682), then what gives? Is the model undersensitive, or missing a flux? Or is the Li

isotope excursionat the PETM caused by something else? It would be good to discuss this, because otherwise it is impossible to tell how much of it is due to missing fluxes/reactions, or to know what they might be.

Our Reply: Modelling studies of the d7Li excursion during the PETM and Cretaceous OAEs found that changes of secondary mineral formation rates in the freshwater system were required to produce the observed d7Li excursions, implying climate-related changes to physical erosion and river transport. cGENIE lacks a complex land-surface model and the representation of slope or soils and hence cannot prognostically simulate changing secondary fractionation with climate. We also deliberately chose to simulate a generic response to carbon release rather than any specific past event for simplicity and to highlight the zero-th order dynamic model behaviour rather than get lost in the weeds of specific observations of any particular past event. However, in line with the inference of Pogge von Strandmann et al. ('The data imply that silicate weathering rates increased fairly dramatically across the PETM. In addition, a shift in the weathering regime to lower intensity (more congruent) weathering . . .'), we can add and test as an illustration, a model parameterization modulates the ïĄd'7Li of runoff inversely to the change in silicate weathering, i.e. higher silicate weathering rates in the model scaling with a reduced weathering fractionation. (This will be provided as an option and as a basis for future model improvements, rather than attempting to encapsulate the entirety of the consequences of secondary clay formation in a single line of code.)

Planned model improvements: A simple optional parameterization that modifies 7Li fractionation associated with weathering, inversely to any climatically induced change in silicate weathering rates (all on a global mean basis).

Planned manuscript improvements: We will point out the discrepancy between our simulation results and the geologic record of transient warming events, highlight the absence of isotopic changes due to erosion rate variations in the discussion of these simulations, but explicitly test and discuss the consequences of assuming a shift to

more congruent weathering associated with increased silicate weathering, following Pogge von Strandmann et al..

Comment 11: In Tables 1-4, values for various parameters and estimates are given, with references for the source in each case. However, from Table 5 on, when talking about model default parameters, there is no such detail. It can be hard then for the reader to know which values are chosen for a good scientific reason, and which are chosen just based on not wanting the model to crash. It would be good to say where these numbers came from. Also it seems like a disproportionately high number of parameters are set to zero- is this because these are the choices made in the chosen scenario only, or are these a lot of parameters that are theoretically included but that people should not use because they may crash the model?

Our Reply: The high degree of parameterization of the four metal cycles means that in every application of the model, the user will have to set the parameters according to their specific experiment design. The experiment protocols provided on GitHub provide two sets of parameter choices for pre-industrial metal cycles. However, we realize that providing generic (often zero) default values was far from helpful, and this we will rectify.

Planned manuscript improvements: To avoid confusion, we will replace the hard-coded default values in the parameter tables with the parameter values we used for our pre-industrial spin-up, and also make it clearer that alternative parameter value choices will be needed for different (paleo) model configurations.

Comment 12: Page 28, Line 26: Would it be hard or computationally-expensive to just scale the Ca2+source at the seafloor to the bottom water temperature that is simulated?

Our Reply: It is certainly possible to extend cGENIE by a dynamic seafloor weathering module. However, there is no well-established formula for the temperature-dependence of seafloor Mg-Ca exchange (Coogan and Gillis [2018]), so any parameterisation choice needs to be justified and evaluated against observational data. Furthermore, one would ideally need to consider seafloor weathering and hence also then need to simulate carbon fluxes to capture the full impact of deep sea temperature variations on the marine carbonate system. Finally, as summarized in Coogan and Gillis [2018], bottom water temperature-driven changes in seafloor weathering may also play a key role in observed variations in 87Sr/86Sr and d7Li through the Cenozoic, meaning that what is ideally required is a comprehensive (and spatially-explicit) representation of fluid-rock reaction in cGENIE. We feel that this would require substantive additional work beyond the scope of our presented model development.

Planned manuscript improvements: We will discuss options to expand cGENIE by temperature-sensitive seafloor weathering in a newly added outlook section.

Comment 13: Section 3.1: I know there are descriptions in the ROKGEM paper, but a cursory explanation of how source rock lithologies are programmed into the model would be helpful in this paper, so the reader doesn't have to read a whole other paper to understand this one.

Planned manuscript improvements: We will extend the description of how weathering fluxes are calculated in ROKGEM as suggested, including the option of using the 2D weathering scheme.

Comment 14: Section 3.7: Should strontium sulphate delivery by Acantharians matter and be con-sidered?

Our Reply: Observations indicate that biogenic strontium sulphate production and consequent dissolution affects the distribution of dissolved Sr, though predominantly in the upper water column due to the relatively high instability of strontium sulphate (e.g. Steiner et al. [2020]). If restricted to a re-partitioning of concentrations in the upper water column (and not significant in terms of a sedimentary sink for Sr), the global budget will be unaffected for all practical purposes. Indeed, existing model-data discrepancies between simulated and measured Sr concentrations are on the scale of ocean basins (e.g. North Atlantic) and we do not think that consideration of Acantharian-driven Sr

cycling will improve the simulation results. However, we will add to the discussion of the simulation of the modern Sr cycle in the model and model-data mismatch.

Planned manuscript improvements: Expanded discussion on the Sr distribution in the modern ocean.

Comment 15: Page 2, Line 5: mantle spelt wrong

We will correct the spelling mistake.

Comment 16: Page 6, Line 13: Perhaps Dellinger et al. (2015, GCA) might be a good paper to include as a citation here?

We will add the reference to the mentioned sentence.

Comment 17: Page 9, Line 9: I would say it's due to analytical inaccuracy, technically, rather than the uncertainty bounds on the values

We will change the expression 'analytical uncertainty' to 'analytical inaccuracy' as suggested.

Comment 18: Page 12, line 21: Would it be possible to give the user a knob to turn to partition more or less hydrothermal flux in one ocean basin vs. another at least? Thinking of the Cretaceous with restricted seaways- Os may be heterogenous, and centres of hydrothermal activity also.

Our Reply: Spatially explicit benthic fluxes of all elements can already be prescribed as boundary conditions to the ocean and we can provide an appropriate reference to the relevant section(s) in the muffin user manual. However, we can go further by adding an optional 'mask' input to constrain hydrothermal fluxes to the masked areas (rather than to the ocean floor globally). While we do not intend to employ the masked hydrothermal flux forcing in this current paper, we can add instructions to do so in the user manual.

Planned model improvements: Addition of an optional spatial mask to restrict hydrothermal input and exchange.

Planned manuscript improvements: We will explicitly state the option to prescribe spatially and temporally heterogeneous benthic metal inputs.

Comment 19: Page 13, Line 19: Why only Li in authigenic carbonates, but not others like Sr?

Our Reply: Our apologies, this sentence was wrong. Li and Sr are incorporated into benthic carbonates (i.e. carbonates forming at the sediment-water interface in reef settings).

Planned manuscript improvements: We will re-word the description.

Comment 20: Page 14, Line 12: buried spelt wrong. Also a reference for the statement would be good.

We will correct this spelling mistake.

Comment 21: Page 18, Table 13: I get that there is no fractionation of Os isotopes parameterized, but this description in this table suggests that the 187/188Os signature of inputs is set to 0, which seems to be wrong..?

Our Reply: Table 13 lists the default values for the Os cycle parameters, which are all set to zero to require the model user to choose a consistent parameter set for their specific experiment design.

Planned manuscript improvements: We will replace the column of default values in these tables with the parameter values we chose to simulate the pre-industrial Os cycle.

Comment 22: Page 19, Lines 21-22: Why is this needed to balance the Ca cycle? And what is the natural process this is supposed to mimic? Can you give a reference?

Our Reply: The primary sink for Mg in the ocean is hydrothermal exchange for other cations (principally calcium) and clay formation (e.g. Coogan and Gillis [2018], Higgins and Schrag [2015]). (To a lesser extent, it also accounts for alkalinity removal

by dolomitisation and the deposition of Mg carbonates, and since the cGENIE only simulates Ca carbonate burial, this exchange term hence helps provide Mg-Ca mass balance.)

Planned manuscript improvements: We will extend our discussion of this set-up and the observed processes in the new manuscript version, including relevant citations.

Comment 23: Page 24, Line 15: not sure 'against' is the right word to use here.

We will replace the word 'against' with the word 'across'.

Comment 24: Page 26, Line 1: no date for the Hall reference

We will add the year 2002 to the reference.

Comment 25: Page 26, Line 11-12: Say 'another' indication that this paper is wrong.. but wasn't the first point of discussion earlier in the paper that these values could be wrong talking about Angino and Billings (1996), not Angino et al.?

Our Reply: The word 'another' is referring to the observation that Sr concentrations in the North Atlantic reported by Angino et al. 1966 are lower than those reported in later studies (e.g. de Villiers 1999) and those simulated (page 24 lines 5-8).

Planned manuscript improvements: We will clarify this in the new manuscript version.

Comment 26: Fig. A1: Never heard of the Indic Ocean! Also see my main gripe. Aesthetics in these plots could also be a lot nicer- it's very default python. For instance the colour palette for the data series or the data point symbols could be used to convey information about ocean basin, or study, or something.. The Os isotope panel for example is really hard to read and not nice on the eyes.

We will split the figure into subfigures for each element and use colour and marker shape to differentiate between ocean basins and studies, as suggested.

Comment 27: Fig. A2-A5: Many of these figures really don't print well. The various

greys and blacks are often very hard to tell apart when printed- the FLUXES vs TUNED for example. I would using some colours to help with this. Fig A2 has a mixture of [] and () in axis titles. Numbers on the axes are unnecessarily small.

We will change the design of our metal-specific model-data comparison plots so that local differences between simulations and observations can be explored in a more visually accessible way.

Comment 28: Fig C1: These datapoints are far too small! Impossible to judge the colour of the points. Ditto the axis labels and legend entries.. There is a missing colour scale label for the bottom left panel, and the X longitude axis is completely cut off for the bottom two panels.

We will increase the size of the data points and make sure all axes are visible.

Comment 29: Acknowledgments: For some reason starts with 'Furthermore'.. Are there some sen-tences missing/lost before?

Our Reply: We thank the reviewer for spotting this confusing sentence start, which was an editing error.

Planned manuscript improvements: We will remove the word 'Furthermore'.

Replies to Reviewer #2

Comment 1: As clearly mentioned and explained in the paper, the cycling of the trace metals (TM) and their isotopic signatures is heavily dependent on the continental weathering pro-cesses, in a way making them difficult to simulate. Here the authors choose to linkthe trace metal fluxes to the weathering flux as calculated by another cGENIE module (ROKGEM). The authors assume that TM fluxes are proportional to the silicate and/or carbonate weathering fluxes. This is probably true for Sr and Ca, but not for Li and Os. Os is heavily dependent on the presence of particulate organic matter. Continental Li fluxes and their isotopic signature are controlled by the interac-tions between secondary phases and the continental waters. Those processes are not

included in the model. Typically, periods of intense weathering will be characterized by a large retention of Li inside secondary minerals, strongly reducing the Li fluxes during the phase of regolith growth, while major cation fluxes released by the weathering of fresh rocks will be at a high level (Vigier and Goddéris, 2014). Li fluxes and isotopic signature are further impacted by the presence of large flooded areas on the continents, where continuous exchanges with clay minerals can occur, directly impacting the isotopic signature of the continental Li discharge (Dellinger et al., 2015; Maffre et al., 2020). I'm not saying that it is mandatory to include all this in a global scale model, but I fear a bit that the proportionality hypothesis will generate wrong interpretations of the seawater isotopic signal. This should be discussed in the paper, for example by stating that this is a first step towards something closer to the physics of the weathering system.

Our reply: We agree that terrestrial processes can substantially alter the composition of dissolved Os and Li in continental run-off and that a representation of these processes in cGENIE would improve the comparability of simulation results with observations and the geologic record. However, a considerably more complex and highly resolved representation of the terrestrial freshwater system and soils would be required to add these processes into cGENIE, which we think is beyond the scope of this manuscript. While our present implementation of the Li cycle in cGENIE has the same caveat as most existing Li cycle models that continental inputs have to be manually manipulated to account for the full impact of climate change on the marine Li reservoir, it still offers new functionalities (i.e. spatially-explicit and dynamic Li burial and the option to simulate Li isotopes alongside other proxies and environmental change in a consistent set-up) to investigate the geologic record. However, as per for our reply to Reviewer #1, we can implement and test (in the context of the abstracted massive carbon release experiments) a simple scheme that inversely couples weathering rate and intensity (and hence inversely relates changes in riverine Li flux to its isotopic composition). With regards to Os – please see our reply to Reviewer #1 and intended explicit spatial test of the importance of spatially-heterogeneous lithological distributions of shale-associated

kerogen.

Planned model improvements: A simple optional parameterization that modifies 7Li fractionation associated with weathering, inversely to any climatically induced change in silicate weathering rates (all on a global mean basis).

Planned manuscript improvements: We will stress that land-surface processes are currently not simulated by cGENIE and discuss consequences for the interpretation of the simulated transient perturbation. We will also discuss options for future improvements of the cGENIE Li cycle in a newly added outlook section.

Comment 2: My second point is related to the residence times of the TM. A clearly stated by the authors, the residence times of the fourth considered TM in the ocean is much longer than the mixing time of the ocean. This implies that the seawater signal will be uniform all around the world. This is shown on figure 1. So why using a complex 3D oceanic model? The mixing will generate a uniform distribution of the TM and their isotopic signature. The interest of including those TM in the model is more to constrain the contribution of the continental and oceanic crust weathering to the flux of elements. If the objective is to constrain the oceanic mixing, elements displaying shorter residence time are best fitted, such as the Nd. This said, the long residence time does not preclude spatial variations, especially on continental margins as it is the case for strontium (El Meknassi et al., Geology, 2020). But those margins are generally not represented in the models. So, I'm just wandering whether a better description of the objectives should appeared or not? (I would say yes).

Our reply: There are a variety of reasons for utilizing a 3D ocean circulation model (although we would argue that it is the least 'complex' in usage and hence most appropriate). For instance: 1. As highlighted by both Reviewers, for some of the TMs, the flux to the ocean can be dependent on the spatial pattern of lithology and surface climate – requiring a 2D land-surface representation and hence at least a '$2\frac{1}{2}$D' (e.g. CLIMBER-2, Brovkin et al. 2012]) if not 3D ocean model component. 2. The ocean sources and

sinks may be spatially heterogeneous, for instance hydrothermal fluxes as highlighted by Reviewer #1 and marine carbonate burial. For Os – if anoxic scavenging is indeed important, a fully 3D representation of the ocean is required in order to simulate the spatial extent and intensity of oxygen minimum zones. 3. The cGENIE model already includes a variety of proxies – published systems such as ïĄď13C, ïĄď34S, ïĄď44Ca, ïĄď56Fe, and I/Cd, plus as-yet unpublished systems such as ïĄď30Si, Cd/Ca – all of which require a spatially-explicit representation of ocean circulation and biogeochemical cycling. The great advantage of adding the TMs is to enable a multi-proxy modelling approach in which e.g. changes in ïĄď13C can be simulated alongside the TMs, which we illustrate in this manuscript in the context of the Earth system response to a massive carbon release. We also respectively disagree with the reviewer that all 4 TMs are uniform throughout the ocean, particularly in the case of Os (as discussed in our reply to Reveiwer #1). (While upper water column Sr concentrations may exhibit distinct non-uniform profiles, we however argue in this case that we need not mechanistically account for this particular heterogeneity.)

Planned manuscript improvements: As suggested, we will state these objectives much more clearly and argue the case at greater length.

Comment 3: My last point is related to the implementation of the runoff. I checked in the Colbourn contribution, but I was not able to understand precisely how it works. A brief description should appear in the text. In summary, this contribution is valuable and should be published, with the above points clarified

Our reply: We are happy to provide an explicit description, including a new figure illustrating the river routing grid, in the revised manuscript. We had also previously devised but not published with, an alternative solute routing scheme that partitions global weathering fluxes according to relative freshwater runoff, rather than drainage basin area. We will also include a description and model test of this modification of the original global mean scheme of Colbourn and co-authors.

Planned manuscript improvements: We will extend the description of weathering and solute routing in our manuscript so that the reader gets a better understanding of the relevant functionalities.

References

Brovkin, V., Ganopolski, A., Archer, D., & Munhoven, G. (2012). Glacial CO2 cycle as a succession of key physical and biogeochemical processes. Climate of the Past, 8, 251-264.

Coogan, L. A., & Gillis, K. M. (2018). Low-temperature alteration of the seafloor: Impacts on ocean chemistry. Annual Review of Earth and Planetary Sciences, 46, 21-45.

Gannoun, A., & Burton, K. W. (2014). High precision osmium elemental and isotope measurements of North Atlantic seawater. Journal of Analytical Atomic Spectrometry, 29(12), 2330-2342.

Higgins, J. A., & Schrag, D. P. (2015). The Mg isotopic composition of Cenozoic seawater–evidence for a link between Mg-clays, seawater Mg/Ca, and climate. Earth and Planetary Science Letters, 416, 73-81.

Lechler, M., von Strandmann, P. A. P., Jenkyns, H. C., Prosser, G., & Parente, M. (2015). Lithium-isotope evidence for enhanced silicate weathering during OAE 1a (Early Aptian Selli event). Earth and Planetary Science Letters, 432, 210-222.

Pogge von Strandmann, P. A., Jenkyns, H. C., & Woodfine, R. G. (2013). Lithium isotope evidence for enhanced weathering during Oceanic Anoxic Event 2. Nature Geoscience, 6(8), 668-672.

Steiner, Z., Sarkar, A., Prakash, S., Vinayachandran, P. N., & Turchyn, A. V. (2020). Dissolved strontium, Sr/Ca ratios, and the abundance of acantharia in the Indian and Southern Oceans. ACS Earth and Space Chemistry.

Tejada, M. L. G., Suzuki, K., Kuroda, J., Coccioni, R., Mahoney, J. J., Ohkouchi, N., ...

& Tatsumi, Y. (2009). Ontong Java Plateau eruption as a trigger for the early Aptian oceanic anoxic event. Geology, 37(9), 855-858.

Woodhouse, O. B., Ravizza, G., Falkner, K. K., Statham, P. J., & Peucker-Ehrenbrink, B. (1999). Osmium in seawater: vertical profiles of concentration and isotopic composition in the eastern Pacific Ocean. Earth and Planetary Science Letters, 173(3), 223-233.

Please also note the supplement to this comment:
https://gmd.copernicus.org/preprints/gmd-2020-233/gmd-2020-233-AC1-supplement.pdf

───────────────────────────────

---

## Author Response (AR2)

Responses to the editor's and reviewer's comments after the second review

We thank the reviewer for their assessment and especially the editor for their careful review of the manuscript. We proofread the manuscript and amended all issues identified by the editor. Below is our point-by-point response to the reviewer's and editor's comments.

Reviewer: I carefully checked the rebuttal provided by Adloff et al., GMD 2020 233). I think the authors have done a careful review of their manuscript, answering part of my questions, and adding more discussions in their revised version for the unsolved questions. I think it can be published as it is now.

Our response: We thank the reviewer for their second assessment and are pleased that our revisions met their expectations.

Editor: Code availability section:
===========================
The DOIs provided to the code archive are only place holders and have to be
replaced by the actual ones. Please notice that we prefer that authors to
give the references to the code in form of a citations
- see https://www.geoscientific-model-development.net/submission.html
- "Prepare your Assets" - "Software and model code". References include
more information than the bare DOIs and may thus be helpful in case the
DOIs have errors. Reference to a code can be included as a BibTeX @Misc
entry, if you use BibTeX. Zenodo allows you to generate such a reference
(please start typing "Copernicus Publications", without the quotes, as the
citation style into the citation style search field, currently in the
right-hand column in the "Share/Cite as" box close to the bottom), then click
on "BibTeX" in the "Export" box below and download the citation, where it is
sufficient to replace the @Software type by @Misc - @Software is not yet
supported by our bibstyle). This way, the citations of your code will be kept
updated on the Zenodo page.

Our response: We obtained a DOI for our model version and added all experiement protocols to this new model release. We updated the code availability section accordingly

Editor: Throughout: "n yrs" should read "nn yr" (units never take a plural 's') or
write out "years".

Our response: We replaced 'yrs' with 'yr' as suggested.

Editor: Throughout: "von Strandmann" should read "Pogge from Strandmann" (the family
name is "Pogge von Strandmann" - one of your co-authors ...)

Our repsonse: We went through all citations and corrected author names.

Editor: p. 2, l. 20: "having become" or "becoming" instead of "have becoming"

Our response: We corrected the grammar as suggested

Editor: p. 2, l. 21: "under under" should read "under"

Our response: We removed the superfluous 'under'

Our response: We spelled out single digits as suggested.

Our response: We corrected this error accordingly.

Our response: We removed the word 'although' and changed the sentence structure to make more sense.

Our response: We removed the 'although' and split the sentence into two.

Our response: We added the missing brackets.

Our response: We replaced 'isotopically heavy Li' with '$^7Li$' as suggested.

Our response: We added the missing 'as'.

Our response: We replaced 'too big' with 'too large'.

Our response: We added labels to all subpanels.

Our response: We reformulated the sentence with the plural for 'reservoir'.

Our response: We corrected this mistake.

The reported $^{88}Sr/^{86}Sr$ is not the value certified for NBS987 which is
8.37861 +/- 0.00325 (see https://www-s.nist.gov/srmors/certificates/987.pdf,
ref.: L. J. Moore, Murphy, T. J., Barnes, I. L., Paulsen, P. J. Absolute
Isotopic Abundance Ratios and Atomic Weight of a Reference Sample of Strontium,
Journal of Research of the National Bureau of Standards, 87(1):1-8, 1982,
doi:10.6028/jres.087.001) The figure cited is, as far as I know, that of the
Eimer and Amend standard (Nier, A.O. The Isotopic Constitution of Strontium,
Barium, Bismuth, Thallium, and Mercury. Phys. Rev. 54(4):275-278, 1938,
doi:10.1103/PhysRev.54.275). 8.375209 is the inverse of 0.1194, i.e.,
the $^{86}Sr/^{88}Sr$ ratio most often used to normalise $^{87}Sr/^{86}Sr$ ratios,
and that 0.1194 is generally said to derive from Nier (1938).
Perhaps unimportant, but nevertheless incorrect and thus confusing.
Pleaseck and correct.

Our response: Thank you for pointing out this wrong label. The 88Sr/86Sr standard in cGENIE is taken from Nier et al. 1938, however the user can change that value. We corrected the reference in the table and mention in the text that this number can be changed.

Editor: Figs. 1, 2, 7, B1:
please notice that we recommend not to use green and red/orange colours in
parallel on a graph (see https://www.geoscientific-model-development.net/submission.html -
"Figures & Tables", point 7)

Our response: We adjusted the colour palettes we used to avoid green and red curves in the same panel.

Editor: Bibliography:
It is sufficient to provide either URL or DOI, the more since both are often
duplicates of each other. As URLs are at the discretion of the publisher and
may thus change at any time, it is best to resort to DOIs.

Our response: We deleted URLs from the reference list.

Editor: p. 49, l. 20: "Earth and planetary science letters" should read "Earth and Planetary Science Letters"

Our response: We corrected the spelling mistake.

Editor: p. 50, ll. 5-6: The DOI for Broecker and Peng is bogus - please discard.
You may provide the following, currently valid, URL:
https://www.ldeo.columbia.edu/~broecker/Home_files/TracersInTheSea_searchable.pdf
Notice, however, the volatility of URLs ...

Our response: We removed the DOI and added the suggested URL instead.

Editor: p. 51, l. 12: "von Strandmann, P. A. P." should read "Pogge von Strandmann, P. A."

Our response: We corrected the name.

Editor: p. 51, ll.15-16: "Reviews in mineralogy and geochemistry" should read "Reviews in Mineralogy and Geochemistry"

Our response: We corrected this spelling mistake.

Editor: p. 51, l. 18: "Geochimica et cosmochimica acta" should read "Geochimica et Cosmochimica Acta"

Our response: We corrected this spelling mistake.

Editor: p. 52, l. 28: in Gehler et al.: "CO2" should have the '2' in index position.

Our response: We corrected this spelling mistake.

Editor: p. 52, l. 35: "my" does not make sense as 'm' would mean 'milli' - use either "My" (best) or "MY"

Our response: We corrected this spelling mistake.

Editor: p. 52, l. 36: "American journal of Science" should read "American Journal of Science"

Our response: We corrected this spelling mistake.

Editor: p. 53, l. 6: "von Strandmann, P. A. P." should read "Pogge von Strandmann, P. A."

Our response: We corrected the name

Editor: p. 54, l. 32: idem

Our response: We corrected the name

Editor: p. 55, l. 3: the '7' of $\delta^7Li$ should be in exponent position

Our response: We put the '7' in the exponent

Editor: p. 55, l. 29: full list of authors required.

Our response: We completed the author list.

Editor: p. 57, l. 12: "von Strandmann, P. A. P." should read "Pogge von Strandmann, P. A."

Our response: We corrected the name.

Editor: p. 58, l. 10: idem

Our response: We corrected the name.

Editor: p. 60, l. 17: "Geochimica et cosmochimica acta" should read "Geochimica et Cosmochimica Acta"

Our response: We corrected this spelling mistake.

Editor: p. 60, ll. 26-27: reference Talley (2002) is incomplete (this is in the Encyclopedia of Global Environmental Change, Volume 1, The Earth System: Physical and Chemical Dimensions of Global Environmental Change) which has editors, a publisher etc. ... please complete.

Our response: We completed the reference.

Our response: We corrected the reference.

---

## Author Response (AR3)

**Response to editor**

We thank the editor for spotting the additional mistakes, and also for his consideration and many efforts throughout the review process!

Here is a point-by-point reply to the technical issues:

Editor: p. 3, l. 8: "three – four" should read "three to four" [my earlier comment was not sufficiently precise - sorry]

Our action: done

Editor: p. 4, l. 8: either delete comma after "values" or add comma before "together"

Our action: We deleted the comma after 'values'.

Editor: p. 5, l. 21: idem

Our action: We deleted the comma after 'values'.

Editor: p. 6., ll. 13-14: check sentence construction: "... with the isotopic composition of Li is expressed as" does not some correct.
Perhaps split in two by deleting "with" and restarting with "The isotopic …"

Our action: We followed the suggestion to split the sentence into two.

Editor: p. 6, l. 27: either delete comma after "values" or add comma before "together"

Our action: We deleted the comma after 'values'.

Editor: p. 8, l. 7: idem

Our action: We deleted the comma after 'values'.

Editor: p. 9 entirely blank ?

Our action: We moved figure 1 to avoid an entirely empty page.

Editor: p. 11, l. 25: "However, n combination" should read "However, in combination"

Our action: done

Editor: p. 14, l. 17: "Becasue" should read "Because"

Our action: done

Editor: p. 16, l. 17: "account" should read "accounted"

Our action: done

Editor: p. 17, ll. 19-20: "although currently this requires" should read "although this currently requires"

Our action: done

Editor: p. 18, ll. 6-7: check sentence(s) construction. Missing closing parenthesis after "(see SI D" ? Spurious parenthesis after "1988)" ?

Our action: We added the missing parenthesis and removed the spurious one.

Editor: Legends to Figs. B4, B5, B7, B8: Salinity does not have any units - please discard PSU.

Our action: done

Editor: Legend to Fig. B3: is the salinity of 34.903495 for the normalisation here still correct? Not sure that salinity can be measured to that degree of precision though.

Our action: We corrected the salinity value to 34.90.